# The Dual Role of Innate Lymphoid and Natural Killer Cells in Cancer. from Phenotype to Single-Cell Transcriptomics, Functions and Clinical Uses

**DOI:** 10.3390/cancers13205042

**Published:** 2021-10-09

**Authors:** Stefania Roma, Laura Carpen, Alessandro Raveane, Francesco Bertolini

**Affiliations:** Laboratory of Hematology-Oncology, European Institute of Oncology IRCCS, 20141 Milan, Italy; stefania.roma@ieo.it (S.R.); laura.carpen@ieo.it (L.C.); alessandro.raveane@ieo.it (A.R.)

**Keywords:** biomarker, blood immune cells, gene profiling, immunomonitoring, immunotherapy, therapy resistance, single-cell RNA sequencing, scRNA-seq, single-cell transcriptomics

## Abstract

**Simple Summary:**

Innate lymphoid cells (ILCs), a family of innate immune cells including natural killers (NKs), play a multitude of roles in first-line cancer control, in escape from immunity and in cancer progression. In this review, we summarize preclinical and clinical data on ILCs and NK cells concerning their phenotype, function and clinical applications in cellular therapy trials. We also describe how single-cell transcriptome sequencing has been used and forecast how it will be used to better understand ILC and NK involvement in cancer control and progression as well as their therapeutic potential.

**Abstract:**

The role of innate lymphoid cells (ILCs), including natural killer (NK) cells, is pivotal in inflammatory modulation and cancer. Natural killer cell activity and count have been demonstrated to be regulated by the expression of activating and inhibitory receptors together with and as a consequence of different stimuli. The great majority of NK cell populations have an anti-tumor activity due to their cytotoxicity, and for this reason have been used for cellular therapies in cancer patients. On the other hand, the recently classified helper ILCs are fundamentally involved in inflammation and they can be either helpful or harmful in cancer development and progression. Tissue niche seems to play an important role in modulating ILC function and conversion, as observed at the transcriptional level. In the past, these cell populations have been classified by the presence of specific cellular receptor markers; more recently, due to the advent of single-cell RNA sequencing (scRNA-seq), it has been possible to also explore them at the transcriptomic level. In this article we review studies on ILC (and NK cell) classification, function and their involvement in cancer. We also summarize the potential application of NK cells in cancer therapy and give an overview of the most recent studies involving ILCs and NKs at scRNA-seq, focusing on cancer. Finally, we provide a resource for those who wish to start single-cell transcriptomic analysis on the context of these innate lymphoid cell populations.

## 1. Introduction

In vertebrates the immune system is an orchestra with two main sections: the innate and the adaptive arms. The role of the orchestra is to defend the host against infections and neoplasia. The two sections have different specializations. The innate section encompasses the complement system (a cascade of plasma proteins that “complements” antibodies against pathogens) and two main families of cells. The two families are (a) the myeloid cells, including mast cells, eosinophils, basophils and phagocytes (neutrophils, macrophages and dendritic cells—DCs), and (b) the lymphoid family, including innate lymphoid cells (ILCs) with its natural killer (NK) cell subfamily [1].

Evolutionary studies suggest that the innate immune system is a very old defense strategy, as it is the dominant one in plants, fungi and insects. It can overcome many threats, but in vertebrates the adaptive section of the immune orchestra is crucial to reach acquired immunity [2].

In the present review, we focus on the role of innate immunity against cancer, and discuss ILC and NK cell phenotype, functions and clinical applications in cellular therapy trials. As single-cell transcriptome investigations are recently offering unprecedented new information on cellular mechanisms, pathways and interactions, we dedicated a chapter to the emerging data from this field. 

## 2. NK and Helper ILCs in Cancer Control, Escape and Progression

The innate immune system has a dual role in cancer since it can either induce tumor regression or favor the spread of cancer through immunosuppression. ILCs are constituents of the innate immune system that have the ability to identify and kill tumor-initiating cells, activating pathways such as antibody-dependent cellular cytotoxicity (ADCC), the complement system and natural cytotoxicity [3,4]. However, ILCs can support tumor progression, inducing the establishment of an immunosuppressive environment by the production of some cytokines [5,6]. ILCs are regulated by many endogenous factors, including cytokines [7], and play fundamental roles in chronic inflammation, metabolic homeostasis, infectious diseases and cancers [8]. Innate lymphoid cells could be either cytotoxic, mainly known as natural killer cells, or helper ILCs. In this section, we will explore the main functions of NKs and ILCs in physiological conditions and in cancer.

### 2.1. Physiological Function of NK Cells and Their Receptors

NK cells represent 5–20% of the lymphocytes circulating in the peripheral blood (PB) and they also populate lymphoid organs and tissues [9,10]. The distinction of NK cells in different subsets is based on the expression of CD16 and CD56 markers. Circulating CD56^bright^ NKs account for around 10–15% of total NKs in the PB. They are characterized by a high production of pro-inflammatory cytokines and low cytotoxic activity [11]. Additionally, they are a source of IFN-γ, TNF-β, IL-10, IL-13 and GM-CSF [12]. NK CD56^bright^ cells react to different stimuli, especially to IL-1β, IL-2, IL-12, IL-15 and IL-18, which are released by both DCs and T cells [13,14]. CD16^+^ NKs (also referred as CD56^dim^) account for around 90% of total circulating NK cells and show a low production of immunomodulatory cytokines upon stimulation; they display antibody-dependent cellular cytotoxicity (ADCC) through the production of lytic molecules such as granzymes and perforin [15,16]. The regulation of CD16 expression onto CD56^dim^ NK is also dependent on metalloproteases that are induced by target cells [17,18]. Furthermore, NK cells undergo CD16 receptor shedding, a mechanism which enhances NK cell survival by avoiding their death due to cell–cell contact with the target cell [19]. 

Another NK cell subset is represented by CD56^dim^ CD16^−^ NK cells [20,21,22]. This subset of NKs, also described as unconventional NKs (uCD56^dim^ NKs), was found to be increased in the PB of patients after hematopoietic stem cell transplantation (HSCT) compared to healthy donors. Unconventional CD56^dim^ NKs were shown to display a high cytotoxicity against hematologic tumor cells in vitro [15]. Recently it has been reported that blocking the inhibitory receptor NKG2A, expressed by this NK cell subset, results in an improvement in NK cell alloreactivity after HSCT, thus showing it as a new potential target for immunotherapy [23].

Natural killer cells play a relevant role in immune responses and in inflammation [24,25,26,27]. For example, they can enter into lymph nodes in inflammatory conditions and regulate T cell activities and responses [28,29]; NKs recognize and kill malignant cells which downregulate the expression of MHC-I in order to hide themselves from T CD8^+^ cells (missing-self hypothesis) [30,31,32].

The two main functions of NK cells concern cytokine production and the secretion of cytotoxic molecules, in which cell–cell contact is required [33]. In order to perceive signals and display their activities, NK cells express activating and inhibitory receptors and, as a consequence of the net result of received signals, they can be activated or restricted [34,35,36,37,38]. 

Natural cytotoxicity receptors (NCRs) are the most frequently expressed receptors on NK cells. NCRs belong to the superfamily of immunoglobulins [31,33], which present a transmembrane domain capable of interacting with signaling adaptor proteins which in turn have immunoreceptor tyrosine-based activation motifs (ITAMs) [39]. In humans the three distinguishable NCRs are NKp46, NKp44 and NKp30 [38,39], and they can recognize ligands from different origins, including viruses, the ones derived from parasites, bacteria and cells [31,39]. The majority of NK cells also present another activating receptor, NKG2D, which binds MHC class I chain-related proteins A and B (MICA and MICB) and UL16-binding proteins. These molecules are either expressed or overexpressed by normal cells under stress conditions and may be present in malignant cells [40,41]. Since NKG2D works as a cell activator, it is responsible for inducing IFN-γ production and NK cell cytotoxicity [40]. Indeed, NKG2D interacts with the costimulatory molecule DAP10, expressed both in humans and in mice, which prompts cell cytotoxicity [41]. Another NK receptor devoted to inducing NK cell cytotoxicity is DNAX accessory molecule-1 (DNAM-1), which binds ligands normally upregulated in cellular stress conditions [38,42,43,44].

The inhibitory receptor NKG2A is endowed with an intracytoplasmic tyrosine-based inhibitory motif (ITIM). Upon ligation this motif is phosphorylated by proto-oncogene tyrosine protein kinase Src and SRC homology 2 (SH2)-domain-containing proteins [45,46]. NKG2A is expressed in approximately half of NKs [47] and performs its function by forming heterodimers with CD94, a protein present on NK cell surfaces. NKG2A/CD94 heterodimers recognize and bind the human leukocyte antigen E (HLA-E) [38,48]. 

Additionally, inhibitory receptors on NK cells are able to block their cytolytic activity when they recognize a self-MHC-I molecule on the target cell [31,48]. Indeed, all the healthy cells have self-MHC-I molecules, which bind to the inhibitory killer immunoglobulin-like receptor (KIR) family on NK cells. In this way, normal cells protect themselves from cytotoxic attack [32,49,50]. Finally, CD94 also forms heterodimers with NKG2C and NKG2E with activation effects.

### 2.2. NK Cell Responses to Cancer

NK cell cytotoxicity is a fundamental property considering its role in anti-tumor responses [51]. The downregulation of NK cell activities in many solid tumors correlates with an increased incidence of metastatic cancer cell spreading [52]. Moreover, in many tumors, such as colorectal cancer (CRC), a low NK cell number increases the risk of cancer recurrence after resection [53]. By strengthening the importance of NK cells in tumor regression, NK cell infiltration in renal cancer correlates with a better prognosis [54]. Another example is that in the PB of patients affected by cervical cancer, NK cells showed a decreased expression of activating receptors, which directly correlated with a higher risk of tumor progression [55]. Patients affected by metastatic cutaneous melanoma who display NK cell infiltration have an improved survival rate, which is amplified in tumors with increased expression of genes related to NK cell stimulation [56]. 

In contrast, NK cell infiltration in non-small-cell lung cancer (NSCLC) does not influence cancer prognosis [57]. A possible explanation of this finding is that in NSCLC there is an enrichment of CD56^bright^ NKs. These cells are localized in the stroma rather than in contact with the tumor cells; thus, they show an increased expression of inhibitory receptors [58]. However, among circulating NK cells, NKp46^+^ NKs have a regulatory role in NSCLC as they may contribute to the formation of an immunosuppressive environment, preventing T cell immunity [59].

A high expression of NKp46 is associated with good survival in patients affected by metastatic melanoma [60], and particular isoforms of NKp30 are correlated with a better response to treatment (for example, with imatinib) [61] in patients affected by gastrointestinal tumors [62]. Moreover, also in NSCLC the expression of NKp46 as well as NKp30 had a predictive role in patient prognosis; in particular, a lower expression of them is associated with a low chance of progression-free survival [63,64]. 

In high-risk neuroblastoma, the NKp30 ligand B7-H6, present on neuroblastoma cells, binds NKp30 and determines NK cell stimulation; however, the soluble form of B7-H6 has the opposite result, increasing metastases and resistance to chemotherapy [65].

In gastric cancer, the expression of NK cell inhibitory and activation markers in a tumor site was similar to normal tissues. However, NK cell functions were impaired, and IFN-γ and TNF-α production was reduced in the tumor microenvironment (TME) [66]. A lower IFN-γ production has also been described in hepatocellular carcinoma, in parallel with a decreased cell cytotoxicity by NK cells, whose regulating mechanisms in this tumor are still under investigation [67].

During breast cancer (BC) progression in tumor-draining lymph nodes, which represent the first site of metastasis, infiltrating NK cells express high levels of inhibitory molecules. For this reason, they might become a potential target for novel immune-therapies [68]. Moreover, both in BC and in pancreatic cancer, NKs have a high expression of NKG2A and reduced expression of activating receptors (NKp30, NKG2D and NKp46), thus resulting in an impairment of NK cell cytotoxicity [69]. In solid tumors, the high expression level of NK cell markers, such as CD56, CD57, NKp30 or NKp46, may predict a favorable prognosis [64]. 

Finally, recent in vitro studies on prostate cancer have shown that tumor-associated NK cells gain the ability to release inflammatory cytokines and chemokines, which in turn support angiogenesis through the stimulation of endothelial cells, recruit monocytes and induce M2-like polarization of macrophages [70]. 

In hematologic malignancies, NK cells have different behaviors depending on the activity of their inhibitory and activating receptors. For example, in myelodysplastic syndrome (MDS) the low expression of the activating marker DNAM1 translates into decreased neoplastic blast killing and concomitantly increased blast infiltration in the bone marrow (BM) [71]. This is similar to what happens with a low expression of NKG2D on NK cells at MDS diagnosis, which associates with a very low cytotoxicity level and contributes to high-risk disease [72]. Furthermore, in acute myeloid leukemia (AML) the low NKG2D expression by NK cells results in an impairment of cell cytotoxicity and leads to IFNγ production [73]. Considering again AML, patients with an elevated expression of the inhibitory molecule NKG2A show a low TNF production and a decreased overall survival rate [74]. Not only in AML but also in chronic myeloid leukemia (CML), the downregulation of NKp30 and NKp46 receptors negatively affects the patient survival rate [75]. In B chronic lymphocytic leukemia (B-CLL) the soluble factor BAG6, which is responsible for apoptosis, protein sorting and transport, differently from its exosomal form, binds NKp30, resulting in NK cell cytotoxicity suppression and CLL immune evasion [76].

In lymphomas, especially in Hodgkin lymphoma (HL) and, to a lesser extent, also in diffuse large B-cell lymphoma (DLBCL) [77], CD56^bright^ NK cells showed a great expression of PD-1 in patients, which suggests a possible mechanism for malignant B cells to inhibit NK cell activity [51]. Similarly, the high expression of PD-1 on NK cells at diagnosis of multiple myeloma is related to a decreased cytotoxicity and a low IFN-γ production [78]. The baseline NK cell count in the PB of patients affected either by follicular lymphoma (FL) or DLBCL, assessed at diagnosis, is known to have a prognostic role on the outcomes of anti-CD20 immunotherapy treatment [79]. In NK cells CD70, the ligand of the tumor necrosis factor receptor CD27, is highly expressed by non-Hodgkin lymphoma (NHL) cells. This ligand expression induces NK cell activation and IFN-γ production, which in turn contributes to NHL cell elimination [80]. CD70/CD27 interactions have become of high interest as possible targets in B cell malignancies. A summary of NK cell roles and functions in cancer is reported in Table 1.

### 2.3. Classification of Helper ILCs and Their Physiological Roles

Helper ILCs are primarily tissue-resident cells which play important roles as mediators in processes such as tissue repair and remodeling, homeostasis and responses to pathogens, allergens and tumors [81]. Helper ILCs can be divided into three different subsets characterized by the expression of specific transcription factors and cell surface markers: ILC1s, ILC2s and ILC3s. These three subsets parallel helper T cell functions [7]; in particular, ILC1s were similar to Th1s, ILC2s to Th2s and ILC3s to Th17s, depending on their cytokine production [82]. Each tissue has a unique ILC subset distribution. For instance, in the small intestine ILC1s are the most predominant subset, while the skin and the lung are populated mostly by ILC2s; in the colon, ILC3s are the most relevant subset [83,84]. In the PB, ILCs are rare, but it is possible to find ILC progenitors (ILCPs), phenotypically similar to ILC3s but able to give rise to ILC1s, ILC2s and NK cells [85]. In humans, ILC1 development is dependent on the transcription factor T-bet. These cells mainly produce Th1-like cytokines, TNFα and IFNγ [86]. The ILC2 subset relies on GATA3 and is characterized by the expression of CRTH2, also known as prostaglandin D2 receptor, IL-33 receptors (ST2) and a variable level of CD117 (also known as c-KIT) [87]. ILC3 development depends on the RORγt transcription factor. This ILC subset expresses CD117 as its cell surface marker [88]. Alterations in ILC number are associated with inflammation and diseases, including cancer (Figure 1). 

### 2.4. ILCs in Cancer

#### 2.4.1. ILC1 in Cancer

The TME affects ILC behavior. Helper ILC frequency, subset distribution and functions may be significantly dysregulated in patients affected by AML. In this disease, hypofunctional ILC1s were increased when compared to healthy donors [90]. Dysregulation in ILC1 functions and number has also been described in CRC [91] and CLL [92]. An increased number of ILC1s was detected in human tissues affected by gastric and esophageal cancer compared to the surrounding healthy tissues [93], as well as in the BM of patients affected by multiple myeloma [94]. Additionally, in patients with myelofibrosis, ILC1s are increased in the PB compared to healthy donors and they show low functional capacity [95]. ILC1s in CLL patients have been shown to be impaired in their function, since after in vitro stimulation they are defective in TNF-α production, in contrast to ILC1s in healthy donors, thus contributing to the formation of an immunosuppressive environment [92]. 

In mice, IL-15 rich environments induced tissue-resident ILC1-like activation and secretion of granzyme B through a mechanism that may involve NKG2D. This receptor is constitutively expressed on these cells; in this case, ILC1s might control tumor progression [96]. 

In melanoma patients the impairment of ILC1 proinflammatory functions in both the PB and lymph nodes infiltrated by neoplastic cells causes the establishment of an immunosuppressive TME [97]. Moreover, in TGF-ß-rich tumors ILC1s can promote tumor growth, as has been shown in vivo in melanoma- or fibrosarcoma-bearing mice [98] (Figure 1). Indeed, TGF-ß is responsible for the conversion of NK cells into ILC1s expressing NKG2A, KLRG1 and other inhibitory receptors [51], thus contributing to tumor growth and progression in an immunosuppressive TME [51]. The conversion of NK into ILC1 has been described as a result of the synergistic effect of TGF-ß and IL-15 involving a class of signaling proteins known as mitogen-activated protein kinases (MAPKs). This, however, does not involve mTOR [99], as has been previously reported [100].

In Crohn’s disease, a bowel inflammatory affection which can be associated with subsequent neoplasia, the percentage of ILC1s is increased [101]. Data from the literature show that these pro-inflammatory ILC1s derive from ILC3 transformation under the influence of TGF-β and IL-23 [102,103], which are abundant in CRC [104]. ILC3s in CRC were impaired in their frequencies and functions [105].

#### 2.4.2. ILC2 in Cancer

A recent study correlated the lack of ILC2s to a higher incidence of tumor growth and easier development of metastases [51]. In fact, the cooperation of ILC2s and DCs results in T cell stimulation and the enhancement of anti-tumor responses [106] (Figure 1). An increased number of ILC2s has been found in human BC tissues when compared to healthy breast tissues [93] and in the urinary immune infiltrates of patients affected by non-muscle-invasive bladder cancer [107].

In mouse models of metastatic melanoma, an anti-tumor activity of ILC2s activated by IL-33 has been shown; in this case, ILC2 cells release IL-5, which in turn recruits eosinophils exerting an anti-tumor activity [108], as also described in CRC [109]. Indeed, the accumulation of eosinophils in some cancer types (such as melanoma and lung cancer) is driven by a signaling pathway mediated by IL-5, whose production from ILCs is properly determined by IL-25 and IL-33 [110].

Another study by Wagner and colleagues demonstrated, through in vivo experiments, that the balance between IL-33/ILC2s/eosinophils and tumor lactate (LA) production is fundamental in regulating tumor growth [111]. Indeed, LA production is related to increased metastatic potential in tumors such as melanoma and it contrasts the immune role exerted by ILC2s [111]. 

In melanoma patients, ILC2 infiltration in the tumor correlated with a good prognosis. However, infiltrating ILC2s in tumors express a high level of PD-1, causing reduced anti-tumor effects; this condition could be overcome through the blockade of PD-1 combined with the administration of IL-33, which favors ILC2 activation [112].

In vivo experiments in pancreatic ductal adenocarcinomas (PDACs) have shown that IL-33 induces the activation of tissue-specific immunity, in particular of tissue resident ILC2s and CD8^+^ T cells, which limits tumor growth. Tissue ILC2s and PD-1^+^ T cells were also found in patients affected by PDACs. As observed in melanoma, ILC2s show a great expression of PD-1, whose antibody-mediated blockade enhances tumor inhibition, representing a new potential strategy for immunotherapy [113].

ILC2 infiltrate in CRC has been described as responsible for tumor burden reduction in vivo and has been found to be associated with improved overall survival in patients affected by this cancer type [114]. Moreover, ILC2s were found to be abundant in CRC-affected tissues compared to adjacent normal ones and to be the main source of IL-9. This cytokine has the ability to activate CD8^+^ T cells, contributing to the inhibition of tumor growth [115].

However, in many cases ILC2s are considered to be harmful, since they produce type 2 cytokines, which promote both tumor onset and progression. A pro-tumorigenic role of ILC2s is the production of amphiregulin (AREG), a strong inhibitor of tumor responses that contributes to the recruitment and activation of regulatory T cells (Tregs) [116,117], also associated with a poor prognosis in breast [118], ovarian [119] and gastric cancers [117]. AREG is a ligand of epidermal growth factor receptor (EGFR), and the interaction between them is responsible for many signaling pathways involving cell proliferation, survival and mobility [120]. 

The recruitment of myeloid-derived suppressor cells (MDSCs) by ILC2s has a detrimental role in tumor surveillance. In the bladder, ILC2s control the local ratio of T cells/MDSCs at tumor sites by recruiting MDSCs through the production of IL-13. This seems to reduce recurrence-free survival in patients [107]. The same cytokine production by ILC2s and the following MDSC recruitment, in the lungs and liver, promotes tissue fibrosis, which might result in cancer [121,122]. In mouse models of lung cancer generated through the injection of tumor cell lines to induce lung tumor lesions, it has been shown that tissue ILC2s increase metastasis and mortality. Indeed, they promote, through IL-5 and IL-33 secretion, the suppression of NK cells, limiting their production of IFN-γ and cytotoxic functions [123].

An investigation on PB in gastric cancer patients has found an upregulation of ILC2-associated cytokines that could in turn favor the formation of an immunosuppressive TME [124]. 

Among hematologic malignancies, ILCs have been studied in particular in acute promyelocytic leukemia (APL) and AML. A study [125] described an increment of ILC2s in the PB of APL patients which were likely recruited by prostaglandin D2 (PGD2) released by APL blasts. The APL blast endogenous ligand B7H6 engages CRTH2 NKp30^+^ ILC2s and contributes to inducing their activation and release of IL-13. These effects result in the induction of monocytic MDSC expansion and the establishment of an immunosuppressive environment. As mentioned above, ILC2s produce IL-9, a cytokine that is responsible for chronic inflammation. In mice, a gene-transfer-induced overexpression of IL-9 promoted lymphoma generation [126]; in humans, high levels of IL-9 are associated with a poor prognosis in HL patients [127]. 

#### 2.4.3. ILC3 in Cancer

The cytokine IL-23 is an important modulator of ILC3 activity in chronic inflammation and plays a role in autoimmunity, host defense and chronic inflammatory diseases. High levels of IL-23 and a high expression of IL-23 receptors (IL-23R) are linked to different human cancers in the skin, breast, stomach and liver [102,104,128].

ILC3s are localized in the gut, and they play a key role in chronic inflammation and gastrointestinal cancer development [129]. Inflammatory bowel diseases (IBD) are characterized by inflammatory conditions in the digestive tract and are associated with an increase in IL-23R signaling, able to promote tumor growth [130]. 

Patients affected by hepatocellular carcinoma have a high level of circulating IL-23, correlating with a poor prognosis. In this kind of tumor and in response to IL-23, ILC3s lack NCR expression and produce IL-17, which in turn limits CD8^+^ T cell anti-tumor activity and contributes to cancer progression [131] (Figure 1). Additionally, in squamous cervical carcinoma, the production of IL-17 by ILC3s correlates with a poor prognosis [132].

ILC3s help the repair process of epithelial tissue through IL-22 production. However, a continuous and uncontrolled secretion of IL-22 may induce excessive inflammation [133]. For example, in non-pathologic conditions, NCR^+^ ILC3s in the intestine produce IL-22 upon interaction with the aryl hydrocarbon receptor (AHR), a ligand-dependent transcription factor [134]. However, an increased amount of intra-tumor levels of IL-22 has been found in CRC and this determines not only an upregulation of anti-apoptotic and pro-proliferating genes [135] but also the induction of abnormal epithelial cell proliferation, as observed in CRC mouse models [136]. Patients affected by CRC who show high levels of circulating IL-22 also show chemoresistance, a possible effect of ILC3 activation [137]. 

In CRC, ILC3s were shown to transdifferentiate into regulatory ILCs (ILCregs) under the stimulation of TGF-β [138]. ILCregs may support cancer progression by means of IL-10 expression, which, in turn, may reduce gut inflammation [139]. On the other hand, IL-10 negatively affects the clinical outcome of many cancer types [140]. In addition, in CRC, during the tumor progression, the amount of NKp44^+^ ILC3s decreases with a concomitant increment in ILC1s and NKp44^−^ ILC3s [141], thus indicating ILC plasticity [101]; the reduction in NKp44^+^ ILC3s is also related to tertiary lymphoid structure reduction in CRC [141]. Another study on CRC reported not only an alteration of ILC3s in terms of frequency and plasticity, but also an imbalance with T cells; ILC3s were shown to be protective against CRC; however, their alteration leads to CRC invasion and resistance to immunotherapies because of microbiota changes [105]. In vivo studies demonstrated that in the gut the production of GM-CSF by ILC3s was driven by macrophages that produce IL-1β in response to microbial signals. This has been found to be important for the maintenance of Treg homeostasis in the colon; indeed, the disturbance of this axis leads to the dysregulation of intestinal immunity, which is related to intestinal diseases [142].GM-CSF has also been found to induce ILC3-mediated acute colitis, driven by IL-23 production by ILC3s, which determines the accumulation of inflammatory monocytes [143]. 

In BC, ILC3s show a pro-tumorigenic behavior and were found to be increased in the TME [144]; indeed, in vitro studies show that ILC3s interact with stromal cells, favoring the expression of RANKL on them and in turn their interaction with RANK is expressed by a breast cancer cell line. This leads to an increment in tumor invasiveness and enhancement—also in correlation with IL-17 production by ILC3 of metastatic spreading in the lymph nodes [145].

ILC3s may originate anaplastic large cell lymphoma, a rare form of NHL. It is the first suggestion of a tumor originating from ILCs [146].

ILC3s might also have a role in cancer control, as described in a mouse model of melanoma. An IL-12 increase in the TME promotes the expansion of NKp46^+^ CD49b-RORγt^+^ ILC3s which, in turn, enhances leukocyte infiltration and tumor suppression [147]. Moreover, in NSCLC, the accumulation of ILC3 NCRs^+^ at the tumor-associated tertiary lymphoid structures has been associated with a better clinical outcome [148].

## 3. NK Cells as Cancer Therapeutics 

This chapter will discuss the current state of NK cell generation/collection and their use in cancer therapy. We will focus on pros and cons related to three major items: (a) the source of NK cells (from progenitors or from already-differentiated cells); (b) the use of autologous vs. allogeneic NK cells; and (c) the use of naïve vs. activated, engineered or monoclonal-antibody-associated NK cells. As recently reviewed by Wendel et al. [2], Riendl et al. [149] and Gauthier et al. [150], clinical immunotherapy trials involving NK cells are ongoing in both hematological malignancies (acute and chronic leukemia, aggressive and indolent lymphomas as well as myelomas) and (to a lower extent) in solid cancers, in the pancreas, the prostate, the ovary, the breast, the brain, the lung, the liver and the digestive tube. 

### 3.1. Clinical-Grade NK Cells Generated from Differentiated or Progenitor Cells

Mature NK cells circulate in limited numbers in adult humans. NK cells do not require an antigen-presenting cell primed to kill target cells. This makes their clinical potential for cellular therapies more prompt [149] compared to T cells, which require days after priming and activation to exploit their anti-cancer activity in vivo.

As reported by Angelo et al. [151], using flow cytometry, when evaluated over time in three individual longitudinal donors, total NK cells comprised a mean of 8.79 ± 3.31, 9.54 ± 1.96 and 5.34 ± 2.23% of all peripheral blood mononuclear cells (PBMCs) across 3–7 longitudinal assessments. In adult donors, apheresis protocols can usually collect PBMC suspensions containing 5–15 % NK cells [152]. Purification strategies using CD3 monoclonal antibodies to deplete T cells and CD56 antibodies to enrich NK cells are then needed to generate clinical-grade, purified (>95%) NK cell suspensions ranging from 5 to 50 × 10^6^ NK cells/recipient kg, even though collections of as many as 10^8^ NK cells/kg have been reported [153].

An NK cell source alternative to the PB of human adults is human umbilical cord blood (UCB), which can be easily and safely collected and is nowadays frozen in large numbers of units for hematopoietic stem cell transplants. UCB contains not only mature NK cells but also NK cell progenitors, which are usually very rare in the PB of adults. These CD34^−^ CD133^−^ CD7^−^ CD45^+^ lineage^−^ cells, progenitors capable of lymphoid/NK cell differentiation, along with slightly more differentiated CD16^+^ CD56^−^ cells, have the potential to differentiate into mature NK cells after ex vivo stimulation with cytokines such as IL-2, IL-15 and/or FLT-3 ligand [154].

Several studies have demonstrated that clinical grade NK cells can be generated from human undifferentiated, purified CD34^+^ progenitors under stromal-cell-free/serum-free medium, heparin and cytokine supplements [155,156]. Finally, there are preclinical data indicating that mature NK cells can be generated from pluripotent stem cells. In some protocols, hematopoietic CD45^+^ CD34^+^ progenitor cells were first generated from embryonic stem cells (ES) or induced pluripotent stem cells (iPS). Subsequently, a 30-day culture on a feeder consisting of a murine fetal-liver-derived stromal cell line and cytokines generated mature CD16^+^ CD56^+^ NK cells with target cell lysis capabilities [157,158]. In another protocol for clinical-grade NK cell generation from ES or iPS cells, an embryonic body assay followed by culture with feeder cells and cytokines was used by Tabatabaei-Zavareh et al. [159] and Knorr et al. [158].

At the present time, a very large majority of clinical trials involving NK cells designed for cancer patients are using mature cells collected from the PB of healthy donors; these cells become naïve or activated after being exposed to cytokines ex vivo or after infusion. Only a minority of trials are using UCB- or PB-derived CD34^+^ progenitors expanded ex vivo before being reinfused, cytokine-induced killer (CIK) cells or engineered CAR-NK cells [2,149], which will be discussed below.

More molecular and functional data on the different activation pathways generated in NK effector cells by these different procedures are now needed to better select which one can be more appropriate for therapeutic purposes in oncology patients.

### 3.2. Autologous vs. Allogeneic Clinical-Grade NK Cells

In the early 1980s, seminal cellular therapy clinical trials for solid cancer patients were based upon crude suspensions of autologous, PB or tumor-infiltrating T and NK cells activated ex vivo by culture in the presence of IL-2 cytokines [160]. In the subsequent decade, some investigators developed protocols to expand autologous NK cells in the presence of IL-2 to treat renal cancer patients [161]. The lack of robust and durable clinical responses suggested that the expression of self-MHC antigens on cancer cells led to inhibitor KIR engagement on autologous NK cells and a lack of cytotoxic activity [162]. In addition, IL-2 is now known to expand Tregs in a loop where NK expansion and functions are inhibited [163].

The first demonstration that allogeneic, KIR-mismatched hematopoietic cell transplantation could generate a strong anti-tumor NK alloreactivity was published in 2002 [164] and paved the way for a large number of clinical trials based upon allogeneic NK administration in non-transplant settings. In most cases, these studies involve conditioning regimens designed to deplete lymphocytes in the recipient [165], and in some others the conditioning of patients is avoided [166]. 

An interesting alternative NK cell source could be “off-the shelf” clinical-grade NK cell lines. The first one to be used in clinical studies was the NK-92 line, administered in some trials for the therapy of patients affected by renal cancer, melanoma or hematological malignancies [167]. NK-92 cells lack CD16 expression and ADCC expression, but are constitutively activated as they lack most inhibitory receptors [168]. On the other hand, NK-92 cells were generated from a cancer patient and therefore needed to be irradiated before infusion, so that their in vivo proliferation potential was limited. In addition, NK-92 cells are highly dependent upon IL-2, which can be toxic when administered repeatedly to patients.

### 3.3. Naïve vs. Activated Engineered or Monoclonal-Antibody-Associated NK Cells

As the cytotoxic potential of naïve, autologous or allogeneic NK cells is somewhat limited, several protocols have been developed to increase NK clinical potential. As we have described above, in several clinical studies NK cells were activated ex vivo or in vivo after IL-2 administration. This cytokine, however, has several side effects and induces an expansion of inhibitory Tregs. To overcome these limits, IL-15 or the IL-15 superagonist ALT-803 have been used more recently to activate NK cells and induce their proliferation in clinical trials [169]. Further preclinical and clinical strategies to activate and expand NK cells by means of IL-12, IL-18, IL-21 or cell feeders expressing one or more of these cytokines in addition to IL-2 have been or are currently investigated in a variety of clinical trials worldwide [2].

A somewhat more complex approach has been developed for the generation of CIK cells. PBMC suspensions were cultured in the presence of IFN-gamma, IL-2, an anti-CD3 monoclonal antibody and IL-1α. This procedure generated T cells with a CD3^+^ CD56^+^ NK-like phenotype along with CD3^−^ CD56^+^ NK cells. These CIK cell suspensions showed a highly proliferative and cytotoxic activity against several types of cancers [170,171,172].

More recently, the addition of IL-15 to CIK-inducing cultures showed an increased killing potential of AML and NHL cells *in vitro* and in clinical trials [173]. CIK cells have been used so far in more than 100 clinical trials, and have shown some encouraging clinical activity in cancer patients [174]. The clinical success of genetically engineered CAR-T cells for the therapy of several hematological malignancies has suggested a similar strategy for NK cells as well. A CAR molecule designed to be transduced to anti-cancer effector cells in order to enhance their activity consists of an extracellular binding domain (usually a single-chain variable fragment or a designed ankyrin repeat protein—DARPin) designed to recognize a neoplastic cell antigen, a hinge region, a transmembrane domain and an intracellular signaling domain [175]. The CAR technology is rapidly evolving, using improved transduction and transfection technologies, novel fragments and adapter molecules, possibly with on/off switch kinetics to increase the clinical efficacy and to blunt the severe side effects, in most cases related to cytokine storms, frequently observed after CAR-T cell infusions [176].

CAR-NK cells have been recently developed for clinical use as an alternative to CAR-T cells. The relatively short lifespan of CAR-NK cells after infusion may prevent the severe cytokine release syndrome frequently observed in CAR-T cell recipients. In contrast to CAR-T cells, CAR-NK cells can kill target neoplastic cells via two separate mechanisms. The first (CAR-dependent antigen recognition) is shared by CAR-T cells, but the second (cytotoxic granules or TRAIL) is NK-restricted. These latter mechanisms might be active even though cancer cells downregulate antigens targeted by CAR molecules, as frequently observed in B cell leukemia or lymphoma patients refractory to (or relapsed after) CAR-T cell therapy targeting human CD19 [176]. In spite of these potential advantages, currently ongoing CAR-NK cell therapy trials (in most cases involving CD19 as a target in NHL and CD123 as a target in AML) are significantly less frequent compared to ongoing CAR-T cell trials. However, this scenario might change in the near future [2]. 

In the past 25 years, the clinical use of monoclonal antibodies for cancer patients has flourished, and this class of drugs is among the most successful in clinical oncology. ADCC is likely the most efficient mechanism of IgG1 clinical activity, and relies on the presence of recipient NK and cytotoxic gamma delta T cells that express Fc gamma IIIa/CD16 receptors [150]. In fact, in Fc gamma RIII knockout mice, ADCC-mediated preclinical activity of monoclonal antibodies is severely impaired [177]. For these reasons, several investigators have developed preclinical and clinical protocols for cancer therapy, associating autologous or allogeneic, fresh or ex vivo expanded/activated NK cells with monoclonal antibodies.

The types of cancer in which these strategies are currently investigated include, among others: (a) multiple myeloma, where autologous NK cells are frequently dysregulated [178] in association with anti-CD138 or -CD319 antibodies; (b) B cell malignancies, in association with anti-CD20 antibodies [179]; (c) digestive tube and lung cancer, in association with anti-EGFR antibodies [180]; and (d) breast cancer, in association with anti-HER2 antibodies [181].

## 4. How Single-Cell RNA Sequencing Studies Have Reshaped the Field and Will Contribute Further

### 4.1. ScRNA Sequencing State of the Art 

Single-cell RNA sequencing (scRNA-seq) is a cutting-edge technology proclaimed as the “Method of the Year” in 2013 [182]. It allows for the dissection of genomic and transcriptomic heterogeneity, revealing mutations, gene expression and structural changes at the resolution of a single cell. 

In particular, scRNA-seq gave the possibility to differentiate among cell populations not distinguishable by cell surface markers and morphology alone. Due to the possibility of identifying novel cellular populations, cellular phenotypes and cellular transitional states (pseudo-time trajectory analysis) [183], this methodology has played a major role in immunology, developmental biology, translational medicine and cancer biology [184,185]. Such information can lead to a much clearer view of the dynamics of tissue and organism development, and on the variation within specialized cell populations so far perceived as phenotypically homogeneous in healthy, stimulated or disease conditions. 

The methods used for single-cell transcriptomic sequencing are divided into droplet-based [186,187] and well-based [188,189,190] platforms and they differ in sensitivity, accuracy, precision and cost. The former has the ability to cover a high number of cells at the expense of gene coverage while the latter does the opposite. The choice of method is determined by the design and aim of the study [191,192]. 

In the last few years, various software suites have been developed on the most frequently used program languages to help dissect the huge amount of information generated by scRNA-seq in a reproducible and rapid manner [191,193,194,195]. Moreover, user-friendly visualization tools, some of which exploit the above-mentioned computational methods, extend the exploration of these data to people without a strong computational background [196,197]. However, the assignment of cell populations previously characterized with surface protein markers and for which a gene expression profile is unknown remains challenging.

In this section, we report a comprehensive summary of the latest studies on the single-cell transcriptomic sequencing analyses of ILCs with a focus on cancer environment and therapy in human and mouse models. We divide the research according to the scRNA-seq characterization of the under-represented helper-like ILCs and NK cells. We provide a table for the identification of ILC subsets (Table 2) based on known gene markers retrieved from the literature of this section. Our aim was to generate an easy-access resource for scientists who plan to characterize ILCs and NK cells at the scRNA-seq level in different tissues and/or conditions.

### 4.2. A scRNA-seq Gaze of Helper-like ILCs

Despite the low frequency of ILCs in the immune cell family, a general agreement on the classification at the transcriptomic level has been reached. Mature ILCs can be classified into three canonical subsets based on their expression of key transcription factors and cytokine production: ILC1, ILC2 and ILC3. However, it is not always possible to distinguish clearly between these populations due to shared gene expression profiles and plasticity. Many intermediate cell clusters have been observed in scRNA-seq data; indeed, ILCPs have also been identified. ILCs play an important role in protective immune responses against intracellular and extracellular pathogens and are central regulators of epithelial barrier integrity and tissue homeostasis. Therefore, the dysregulation of their function contributes to the development and progression of multiple inflammatory diseases [230].

ILCs are involved in cancer immunity and can contribute to tumor-associated inflammation. Their role appears to be TME-dependent [203]. For example, the scRNA-seq profile of intratumoral ILCs has been studied during CRC progression in mouse models [138]. At a late stage of tumor progression, a new regulatory population of tumor-infiltrating ILCs, named ILCregs, has been identified. It contains high levels of the fate decision factor gene *Id3* [139] and produces IL-10, which negatively modulates inflammatory responses. At the same stage, ILC3s decrease with the increase in this cell population, demonstrating a conversion of ILC3s into ILCregs over CRC tumor progression. Moreover, ILCregs do not express the conventional signature markers of other ILCs but have elevated levels of TGF-β signaling genes [138]. In humans, the single-cell transcriptome characterization of ILCs derived from healthy blood, normal mucosa, and CRC tissue revealed that the normal gut mucosa contains ILC1s, ILC3s and ILCs/NKs, but no ILC2s cells, thus confirming their important involvement in inflammatory conditions in this tissue [230]. In contrast to the control tissues, ILCs from CRC patients were found to contain two additional subsets: a CRC tissue-specific ILC1-like subset and an ILC2 subset, the only two clusters that differ from the normal gut. Moreover, the expression of the *SLAMF1* (signaling lymphocytic activation molecule family member 1) gene was upregulated in intratumoral ILCs, but only weakly expressed in their healthy counterparts, suggesting that *SLAMF1* expression can be a predictive biomarker in CRC [203]. Finally, CRC severity was associated with a large number of ILC1s and an abnormal lower level of ILC3s in the intestine than at steady state [204].

A different contribution of ILC3s, isolated from ceca and mesenteric lymph nodes, has been observed in *Salmonella*-infected mice [201]. ScRNA-seq analysis revealed that ILC3s are the most abundant subset in the two tissues analyzed from wild-type (*wt*) infected mice, followed by NK cells, ILC1s and ILC2s. On the other hand, RORα-depleted mice resulted in a strong increase in NK cells, whereas the frequency of ILC3s is highly reduced and ILC2s are ablated. These data implied that RORα is required for ILC2 development and lead to significantly impaired maintenance of ILC3s that lacked features of cytotoxicity (low expression of *Ifng* gene) with a concomitant expansion of cytotoxic NK cells that attenuate inflammation in the gut. The above-mentioned studies present different behaviors of ILC3s during tumor progression and infection, highlighting their possible roles in other diseases. Another example is given by a decrease in ILC3 in human and mouse psoriatic skin in response to anti-TNF antibodies [231]. In turn, this suggests that ILC3s may contribute to pathogenesis [232], with a role different from that reported in cancer [138]. Moreover, the transcriptional profiles of skin ILCs, as stated by Bielecki et al. [232], showed the presence of highly heterogeneous ILC3-like cells, which are induced in a time-dependent manner, reflecting diverse origins, such as quiescent cells, ILC2s or from a group of cells characterized by a dense transcription continuum (“clouds”).

ILC3 conversion to an ILC1-like phenotype was also observed in the human tonsil [103]. This study was pivotal in emphasizing the relevance of the tissue niche in creating a microenvironment that promotes ILC diversity and functional adaptation to local stimuli. Finally, different organs present divergent proportions of ILC subsets; for example, in healthy human colonic mucosa the most abundant ILC subtype is ILC3, and it showed the highest degree of heterogeneity when compared to other human compartments [204].

The characterization of ILCPs, at the scRNA-seq level, has been obtained for the first time by the analysis of data derived from sorted ILCs from mouse BM [199]. Different ILC precursor subsets have been recognized and the relevance of this discovery is that the target for many cancer immunotherapies, *PD-1*, was identified as one of the most expressed genes by a cluster of ILCPs [233]. At a later point, another study [198], using a droplet-based scRNA-seq method, aimed to dissect the ILCPs. This research identified two stages of BM ILC development as distinct clusters among the landscape of ILCPs were observed. These populations were named “specific” early innate lymphoid progenitors (EILPs) and “committed” EILPs. They presented different transcriptional profiles, and the gene *Tcf7* (coding for the Tcf1 protein) plays a major role in their differentiation. *Tcf7*, together with *Il18r1*, were observed to be highly expressed in progenitor or immature ILCs in the lungs [207].

Zeis et al. [207] proved that the full spectrum of ILC2s in infected mouse lungs is generated by cells derived from BM, and confirmed that the ILC phenotypes are strongly derived from local tissue niches rather than from progenitor origin [207]. 

Many studies have characterized ILC2s in mouse lungs [203,210,211,212]: recent research [206] analyzed both adult and neonatal lungs and found two separated clusters of ILC2s defined by different gene signatures, one expressing conventional ILC2 genes such as *Il1rl1* and *Tnfrsf18* and the other expressing *Cd7*, *Runx3*, *Tcf7* and *Il18r1*, but not *Il1rl1* and *Tnfrsf18*.

Tissue-specific subclasses of hepatobiliary ILCs were also evaluated in the liver and the extrahepatic liver duct (EHLD), after IL-33 administration [200]. The analyses revealed a canonical ILC2 and a liver-specific ILC2 cluster. Moreover, a bile-duct-specific class of ILC1s, restricted only to EHBD, which expresses transcripts consistent with the ILC1 subpopulation, and an undescribed class of biliary immature myeloid (BIM) cells, which expresses genes of immature myeloid cells, were found for the first time. 

In mice infected by *Nippostrongylus brasiliensis,* ILC2s extracted from lung tissue presented two clusters. The “natural” cluster expressed low levels of the *Klgr1* gene, while the second cluster, named “inflammatory”, had high levels of this gene. 

Beside the presence of ILC2s in mouse mucosal lung tissue, these populations were also to be involved in many physiological and pathological processes in skin and the gastrointestinal tract, as investigated at the transcriptional level in [234]. In contrast, healthy colon tissue from humans was demonstrated to be ablated by ILC2s at the transcriptome level [204]. Different human dermal ILCs revealed two clusters of ILC2s with different gene expression [205]. One cluster, characterized by a high expression of the *CCR10* gene, was identified as the least mature, and was composed of a lower number of cells. While the other, in which most of the ILC2s fall, was represented by a high expression of *KLRG1,* a key surface marker of ILCs. Therefore, these results revealed human ILC2 subsets with potential inflammatory profiles. 

ILCs retrieved from tonsil, blood and other organs in humans were also investigated by Mazzurana et al. [204]. This study provided important gene markers for circulating and tissue-resident naïve human ILCs. Signs of both recirculation and tissue residency have also been observed in colons and lungs. Putative ILC1s, with some gene signatures typical of T cells, were reported in blood and tonsil. 

### 4.3. NKs under a Single-Cell RNA Microscope 

At the single-cell RNA level, NK cells have been widely characterized in a variety of tissues, both in human and mouse models. In particular, it has been demonstrated that circulating and tissue-resident NK cells are rewired in pathological conditions [218]. 

In the last few years, different studies characterized murine and human circulating (PB) as well as tissue-resident (spleen, bone marrow - BM and liver) NK cell populations at the scRNA-seq level [209,215,221]. Two pivotal studies identified two NK subsets (clusters) in the PB of both species and three and four in mouse and human spleens, respectively (four were also the clusters identified in human BM). Their ubiquitous location confirmed their origin in BM and the subsequent spread, through the bloodstream, towards different organs [235]. Transcriptional profiles of these cells revealed two clusters, shared between tissues and species, associated with CD56^dim^ and CD56^bright^ populations and named NK1 and NK2, respectively. The NK1 cluster expresses a gene signature related to cytotoxicity, while NK2 is enriched in genes related to the production of cytokines and chemokines. In addition to these clusters, previously uncharacterized spleen-specific NKs were observed. These cells resemble, at the transcriptomic level, the NK1 or NK2 clusters mentioned above. Furthermore, a progenitor NK cluster (NK0) was exclusively identified in human BM, and it presents a gene signature similar to NK CD56^bright^ [215]. Pseudotime scRNA-seq analyses of circulating NKs showed a previously unknown transient cellular state (cluster) between NK1 and NK2, which are placed at the end of the two branches on the pseudotime trajectory. This new NK cluster provides the first transcriptional evidence that there is a CD56^bright^ to CD56^dim^ NK transition, even if the two cell populations are not developmentally related, in a human in vivo model [215]. Finally, a novel classification, with the aim of discriminating circulating and liver-resident NKs, was proposed on the basis of the expression of chemokine receptors [221].

The role of NKs in cancers has been widely treated in the introduction of this review, and usually a general increase in NK cells, but in particular of the cytotoxic CD56^dim^ population, is associated with a better prognosis and cancer regression (see Section 2.2, NK Cell Responses to Cancer). However, a recent scRNA-seq study analyzed the immune system of the cancer-resistant naked mole and found a lack of canonical NK cells in this rodent. The concomitant increase in several myeloid cell populations (which were not found in control mice), suggested an alternative innate immune response able to contrast bacterial infection and/or cancer progression [236]. 

Specific NK cell programs within the TME in breast cancer (BC) and the melanoma of mouse models have been identified using the scRNA-seq technique [210]. In particular, homogeneous and robust groups (metacells) of intratumor NK cells show a decrement in the expression of genes found in mature effector NK cells (i.e, *Itgam*) at the expense of an increase in genes characterizing immature NK cells and ILC1 populations, such as *Itga1* (coding for CD49a). The authors associated this response with a replacement of exhausted NK cells with circulating and more active NKs. They found a reduction in non-tissue circulating NKs, as if a reprogramming took place to drive the NK cells toward the TME. 

Other genes coding for glycolysis enzymes (*Aldoa, Tpi1, Pgk1,* Pkm2 and *Ldha*) have been found to be upregulated on NK cells of BC and lymphoma mouse models [210,213], reflecting an alteration in the function of NKs in a hypoxic TME [237]. Interestingly, mice carrying lymphoma and deficient for a gene (*Hif1a*) coding for a transcription factor useful for hypoxia adaptation also show reduced tumor growth in association with an increase in activated tumor-infiltrating NK cells (high expression of *Ifng*, *Cd69, Prf1, Gzma* and *Gzmb)* [213]. These cells also show an increased expression of genes related to the nuclear factor kappa B zeta (NF-κB) pathway. The NF-κB pathway modulates innate immunity [238] by means of IL-18 produced by myeloid cells in the TME [213]. Therefore, these results, together with an increase in the survival in lymphoma patients with an *NK-IL18-IFNG^hi^* signature, pave the way to potential cancer therapy involving the HIF-1α gene [213]. A general increase in intratumor NK cells was also observed when targeted therapy (ceritinib–trametinib or dabrafenib–trametinib) was administered after immunotherapy (αPD-1) in mouse melanoma models [239]; however, the activation profiles of these NK subpopulations has not been tested.

In humans, scRNA-seq has been exploited to observe NK cell variations at the BM level in patients with AML [215] and mantle cell lymphoma (MCL) [216]. In the former research, Crinier et al. [215] identified BM NKs in AML patients that were enriched in genes involved in responses to cytokine and type I interferon signaling pathways. Healthy donors had BM NK cells with a more cytotoxic profile, confirming a functional impairment of NK cells during AML [240]. More than one hundred genes were found to be upregulated in AML patients, and among them was *CD160*, coding for an NK cell activating receptor [241], is correlated with a better survival of these patients. Further investigations on the role of this marker on NKs during AML could be relevant for cancer therapy or for the characterization of this disease [215]. In the other above-mentioned study, involving only one patient affected by MCL, a decrease in NK cell populations was associated with B cell activity. This research would need further validation, mainly involving an increase in the sample number [216].

Using two different technologies (droplet- and well-based methods), scRNA-seq analyses of stromal cells from CRC patients revealed a large subset of tumor-resident NKs [228]. This demonstrates once again that NK cells are very heterogeneous in the TME and may have different roles. These cells contained a set of genes coding for inhibitory receptors as well as cytotoxic and proliferative molecules, similar to the gene signature found in CD8^+^ exhausted T cells described previously in CRC [242].

Malignant cells isolated from the ascites of patients with ovarian cancer [223,224] were enriched in NK clusters presenting high levels of the *CCL5* gene, whose expression was deficient in ovarian cancer cells that expressed high levels of *SDC4*. The main finding of this study was the correlation of *CCL5* gene expression in NKs with ovarian cancer progression and poor prognosis. This led to a potential role of NK cells in anoikis, a negative cell regulation of programmed cell death, resistance in the TME of ascites [223]. 

Functional state analyses of NK cells in human melanoma also identify highly cytotoxic clusters with a high expression of chemokine genes *CCL3*, *CCL4*, *CCL4L2* and *CCL5*. These chemokines bind to their receptors and play a critical role in the recruitment of T cells and other immune cells [243]. Other cells with low levels of these transcripts were identified as poorly cytotoxic [214]. This demonstrates that NK cells are very heterogeneous in the TME and can have different roles. In the same study (also including PB cells from melanoma patients) [214], the patterns of gene expression found within NK clusters were similar to those identified by Crinier et al. [209].

Very recently, the scRNA-seq of primary tumors, lymph nodes and PBMCs in human head and neck squamous cell carcinoma (HNSCC) patients aimed to reveal the innate immune landscape of this pathology [229]. The analyses, after an unbiased cell assignment of the scRNA-seq clusters, reveal two intratumor NK cell populations with peculiar characteristics. The first had a high anti-tumor activity and resembled intraepithelial ILC1s (a high expression of genes coding for granzyme and perforin). The other had low capacity to control tumor growth and expressed *CD49a* and *NR4A2*.

Since NK cells are one of the main actors in the innate immune system, analyses of their transcriptome during infection could unveil functions and peculiar mechanisms also found in cancer conditions. Recently, two studies focused on the analysis of ILCs during an infection of *Toxoplasma gondii* in mice [211,212]. In the liver, one study identified a shift of NK cells into two distinct ILC1-like cell subpopulations appearing only during the infection in *wt* mice [212]. The other identified an upregulation of effector genes and proliferation markers (*Ifng*, *Gzmb* and *Mki67*) in NK cells isolated from peritoneal exudates at different stages of the infection. Moreover, a peritoneal cell cluster of early activated NK cells with a high expression of genes induced by cytokine stimulation was identified [211]. These findings suggest that *T. gondii* infection rewires NK cells and ILC1s, inducing permanent changes.

Lately, due to the global pandemic of COVID-19, many studies widely dissected the immune landscape in humans during viral infection, and some of them identified changes in NK cells [217,225,226,227]. A decrease in NK cells was associated with the severity of the condition in the PBMCs of COVID-19 patients [227]. Specifically, a reduction in CD56^dim^ NKs was recorded in COVID patients with acute respiratory distress syndrome, and a depletion of CD56^bright^ NKs was encountered in all COVID-19 samples.

## 5. Conclusions and Future Directions

Although ILCs and NK cells clearly play a cornerstone role in cancer control, our knowledge of their specific contribution is still limited. One main reason for this is the lack of understanding of the relationship between their phenotype and their functions. In this context, scRNA-seq applications in cancer research have followed one another with crucial new findings [244]. The extension to diagnostic and clinical settings might suggest new therapeutic options. We highlighted the latest discoveries of potential therapeutic targets and diagnosis markers in rare cell populations that would not have been possible to identify without the advancements in the field of single-cell transcriptomics. The identification of unknown and cancer-resistant cell populations after tumor removal and the fine-scale characterization of the immune TME may give new insights into the diagnosis and treatment of these malignancies, leading to a personalized cancer therapy. 

Furthermore, the continuous development of single-cell transcriptome technology and its combination with multi-domain technologies will offer new insights into next-generation sequencing [245]. Its unique role in the field of cancer will help to explain the biological mechanisms that could not be understood before and promote the development of personalized medicine, bringing new breakthroughs in clinical diagnosis, treatment and prognosis of patients. This will require a combined effort of clinicians and computational as well as laboratory scientists. With this intention, we provide here, for the first time, a summary of gene markers used to classify ILC populations (including NKs) at the scRNA-seq level, which may be useful for researchers who intend to start new transcriptomic studies on ILCs (Table 2).

## Figures and Tables

**Figure 1 cancers-13-05042-f001:**
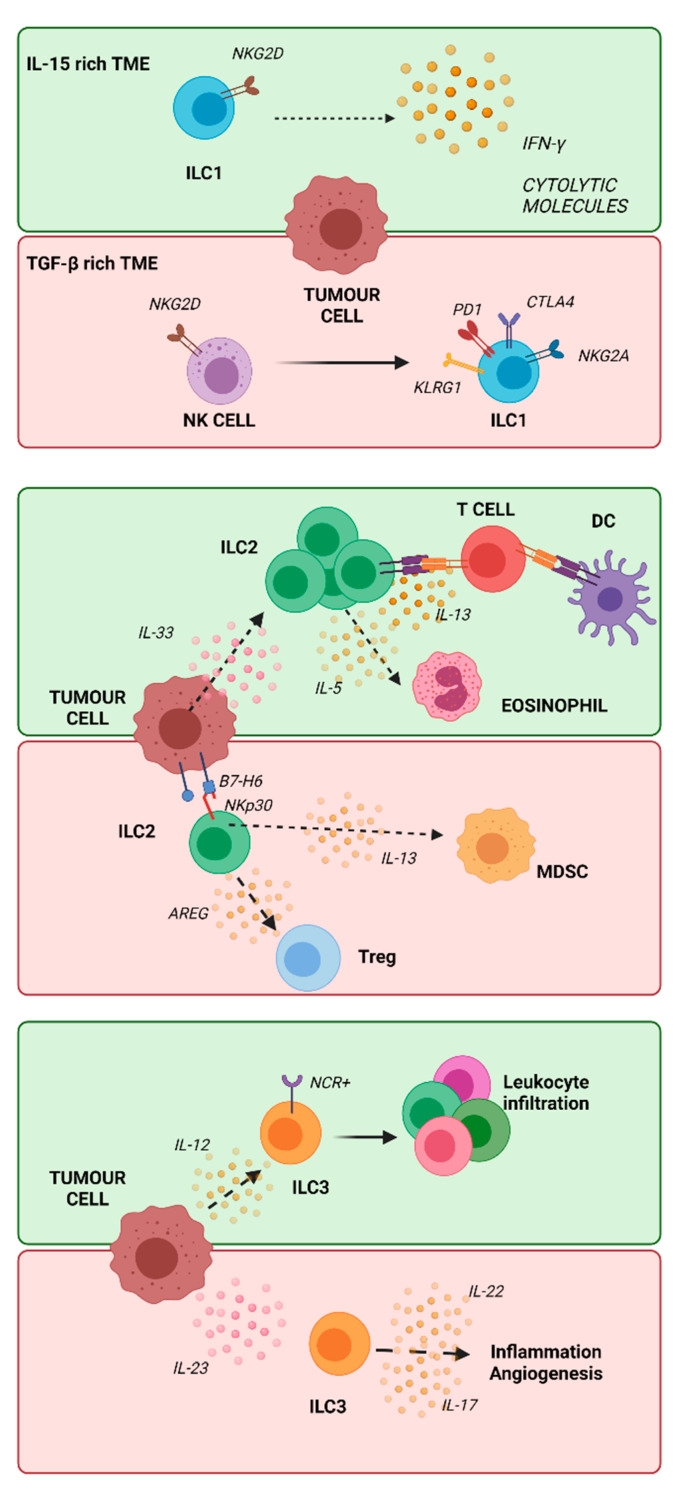
Helper ILCs and their role in cancer development/regression. The illustration summarizes the different roles of helper ILCs in cancer. In green, it has been reported how ILC1s, ILC2s or ILC3s could be helpful in terms of cancer control; in red, how they can be harmful and support cancer progression or promotion. Some of the pathways represented in this illustration were described in Crinier et al., 2019 [89]. In an IL-15-rich TME, ILC1s can produce lytic molecules, contributing to tumor cell elimination. However, TGF-β causes a transition from NK cells to ILC1s, which are less cytotoxic; this mechanism may sustain tumor growth/progression. Considering ILC2s, they can help tumor regression by recruiting T cells and eosinophils; on the other side, through IL-13 production, they can recruit MDSCs and through AREG and Tregs they contribute to the establishment of an immune-suppressive environment. ILC3s can contribute to tumor regression by inducing leukocyte infiltration through IL-12 release; on the other hand, they can exert a pro-tumor role through the production of IL-17/IL-22 under the influence of IL-23 produced by the tumor. (Created with BioRender.com).

**Table 1 cancers-13-05042-t001:** NK roles and phenotypes in different types of cancer: NSCLC (non-small-cell lung cancer), BC (breast cancer), CRC (colorectal cancer), AML (acute myeloid leukemia), CML (chronic myeloid leukemia) and HL (Hodgkin lymphoma).

Type of Cancer	NK Cell Role or Phenotype	Impact on Tumor	References
NSCLC	Low cytotoxicity of CD16^+^ and CD56^+^ NKs NKp46 expression	Not known Immune-suppressive environment	[57] [59]
BC	NK infiltrate with high expression of NKG2A, low expression of Nkp30, Nkp46 and NKG2D	Impaired cytotoxicity	[68,69]
Gastric cancer	NK with lower cytotoxicity	Low TNF-α and IFN-γ	[66]
CRC	Low NK number	Association with cancer recurrence after resection	[53]
Renal cancer	Tumor-infiltrating NK	Better prognosis	[54]
Cervical cancer	NK cells with low activating receptor expression	Contributes to cancer progression	[55]
Metastatic cutaneous melanoma	High infiltration of NK cells	Better prognosis and survival	[56]
Hepatocellular carcinoma	High expression NKG2A, low expression of Nkp30, NKG2D andNkp46	Impairment of cell cytotoxicity	[67]
Myelodysplastic syndrome	Low expression of DNAM-1	Lower blast killing, higher blast infiltration; high-risk disease	[71]
AML	Low NKG2D High NKG2A	Impairment of cell cytotoxicity, low IFN-γ production	[73]
CML	Downregulated Nkp30 and Nkp46	Impaired survival	[75]
HL	PD-1 expression	Immune evasion	[77]
Multiple myeloma	PD-1 expression	Not known	[78]

**Table 2 cancers-13-05042-t002:** ILC/NK classification using scRNA-seq analyses: gene markers of both humans and mice are reported. Dashed lines and upper/lower-case characters separate the genes markers of the two species.

Cell Populations	Specific Cluster Name ^1^	Organism	Tissue	Cell Source	Gene Marker ^2^	Reference
ILCP	ALP	Mouse	BM of *Tcf7^E^*^GFP^ mice	Lin^−^cKit^+^2B4^+^α4β7^−^Flt3hiIL-7Rα^+^	*Tcf7 Flt3 H2dma Sp1 Mef2c*	[198]
	sEILP		BM of *Tcf7*^EGFP^ mice	Lin^−^cKit^+^2B4^+^α4β7^+^Tcf7-GFP^+^	*Tcf7 Irf8 Runx3*	[198]
	cEILP		BM of *Tcf7*^EGFP^mice	Lin^−^cKit^+^2B4^+^α4β7^+^Tcf7-GFP^+^	*Tcf7 Runx3 Tox*	[198]
	General		BM of *Tcf7*^EGFP^ mice	Lin^−^cKit^+^2B4^+^α4β7^+^Tcf7-GFP^+^	*Zbtb16 Lmo4 Rora*	[198]
	General		BM	Lin^−^Flt3lo/^−^IL^−^7Rαlo/^+^α4β7^+^	*Zbtb16 Il7r Kit Itga4 Itgb7 but low Myc*	[199]
	General		BM	Lin^−^Flt3lo/^−^IL^−^7Rαlo/^+^α4β7^+^	*Tcf7 Il18r1*	[199]
ILC1	General	Mouse	Colon-tumor-infiltrating ILCs	CD45^+^Lin^−^CD127^+^	*Xcl1 Tbx21 Klrb1c Ncr1 Ifng Klrd Klrk1 Klrc1 Ctsw*	[138]
	ILC1s and/or cNKs		BM	Lin^−^Flt3lo/^−^IL−7Rαlo/^+^α4β7^+^	*Cish*	[199]
	Bile duct-ILC1		Liver or EHDB after Il-33	CD45^+^Lin^−^	*Cd28 Il12rb2 Thy1 Cd93 Tbx21 Ifngr1 Il2 Il10*	[200]
	General		*Salmonella*-infected ceca of WT or Rora^sg/sg^ BM-transplanted (BMT) chimeric mice	CD45.2^+^Lin^−^CD90^+^	*Xcl1*	[201]
	General	Human	Tonsil	Lin^−^CD127^+^NKG2A^−^CD16^−^CD117^−^CRTH2^−^	*CXCR3 IFNG*	[202]
	General		CRC tissue	Lin^−^CD45^+^CD127^+^	*CD3D CD3G CCL4 IFNG IKZF3 PRDM1*	[203]
	General		Lung, blood, colon and tonsil	Lin^−^CD45^+^CD127^+^CD117^−^CRTH2^−^	*IL7R*	[204]
	General		PB	Lin^−^CD45^+^CD127^+^CD117^−^CRTH2^−^	*ETS1 TBX21 EOMES IFNG BCL11BTCF7*	[205]
ILC2	General	Mouse	BM	Lin^−^Flt3lo/^−^IL-7Rαlo/^+^α4β7^+^	*Pdcd1 Bcl11b Icos Rora Gata3*	[199]
	General (from clusters c8 to c10)		BM	Lin^−^Flt3lo/^−^IL-7Rαlo/^+^α4β7^+^	*Il2ra Il1rl1 Bmp7 Pparg*	[199]
	General (cluster c8)		BM	Lin^−^Flt3lo/^−^IL-7Rαlo/^+^α4β7^+^	*Itgb3 Pbxip1 1700113H08Rik*	[199]
	General (cluster c9)		BM	Lin^−^Flt3lo/^−^IL-7Rαlo/^+^α4β7^+^	*Ccr8 Gclc*	[199]
	General		Colon tumor infiltrating ILCs	Lin^−^CD45^+^CD127^+^	*Gata3 Il4 Il5 Klrg1 Il1rl1 Il13 Il17rb Fosb Hes1 Itga4*	[138]
	Liver-specific ILC2		Liver or EHDB after Il-33	CD45^+^Lin^−^	*Ccr4 Tnfsf9 Il12rb1 IL10 NFil3 Idi1 IL9r*	[200]
	Canonical ILC2		Liver or EHDB after Il-33	CD45^+^Lin^−^	*Rora Gata3 Arg1 Ccr8 Klrg1 Icos Il17Rb Ccr2 Il5 Il13*	[200]
	General		Adult and neonatal lung	CD45lo/^+^Lin^lo^ RORγt^−^YFP^−^YFP^+^	*Areg Il7r Rora and Gata3 and the lack of expression of stromal NK/ILC1 B and T cell genes*	[206]
	Conventional ILC2		Adult and neonatal lung	CD45lo/^+^Lin^lo^ RORγt^−^YFP^−^YFP^+^	*Il1rl1 Tnfrsf18 Areg and Arg1*	[206]
	General		Adult and neonatal lung	CD45lo/^+^Lin^lo^RORγt^−^YFP^−^and YFP^+^	*Cd7 Runx3 Cd2 Tcf7 Il18r1*	[206]
	ILC2 mature		Infected lung	(Lin^−^) CD45^+^Il7ra^+^Thy1^+^/lowNK1.1^−^	*Gata3 Il1rl1 Klrg1 Bcl11b*	[207]
	ILC2 effector		Infected lung	(Lin^−^) CD45^+^Il7ra^+^Thy1^+^/lowNK1.1^−^	*Il5 Il13 Il17a*	[207]
	Natural ILC2		Lung	Lin^−^CD3^−^TCRβ^−^Thy1^+^	*Cd69 Nfkbid Fos Il13 Il1r2 Cxcl3 Calca Cxcl2 Il5 Csf2 Ccl1*	[208]
	Inflammatory ILC2		Lung	Lin^−^CD3^−^TCRβ^−^Thy1^+^	*Gzma Lgals1 Stmn1 Ccr9 Lgals3 Klrg1*	[208]
	General		*Salmonella*-infected ceca of WT or Rora^sg/sg^ BM-transplanted (BMT) chimeric mice	CD45.2^+^Lin2CD90^+^	*Gata3 Il4 Il17rb Il1rl1 Rora Il7r*	[201]
	General	Human	Lung, blood, colon and tonsil	CD3^−^CD4^−^Lin^−^CD45^+^CD127^+^CD117^+^/^−^CRTH2^+^	*IL7R GATA3 MAF PTGRD2 HPGDS*	[204]
	c-kit- ILC2		Skin	CD45^+^Lin^−^CD127^+^CRTH2^+^CD117^−^/^+^	*GATA3 CRLF2 IL17RB CCR6 RORA BCL11B*	[205]
	c-kit+ ILC2		Skin	CD45^+^Lin^−^CD127^+^CRTH2^+^CD117^−^/^+^	*CRLF2 IL17RB CCR6 RORA BCL11B ZBTB16 RORC*	[205]
	General		Tonsil	Lin^−^CD127^+^NKG2A^−^CD16^−^CRTH2^+^	*GATA3 IL1RL1 IL17RB PTGDR2*	[202]
ILC3	General	Mouse	BM	Lin^−^Flt3lo/^−^IL-7Rαlo/^+^α4β7^+^	*Tbx21 Il2rb Ncr1 Cxcr3 Ctsw*	[199]
	General		Colon-tumor-infiltrating ILCs	Lin^−^CD45^+^CD127^+^	*Rorc Il22 Ncr1 Sepp1 Ccr7 Fcer1g Cd74*	[138]
	ILCreg		Colon-tumor-infiltrating ILCs	Lin^−^CD45^+^CD127^+^	*Id3-Il10-Ctla4-Klf2-Tnfrsf18-Tnfrs8-Tnfrs9*	[138]
	General		*Salmonella*-infected ceca of WT or Rora^sg/sg^ BM-transplanted (BMT) chimeric mice	CD45.2^+^Lin2CD90^+^	*Rora Il7r Thy1 Gzmb Xcl1 Ncr1 Ifng Cxcr6 Cd4*	[201]
	General	Human	Lung, blood, colon and tonsil	CD45^+^Lin^−^CD127^+^CD117^+^CRTH2^−^	*KIT IL1R1 IL23R RORC*	[204]
	General		Tonsil	expression of NKp44 CD62L HLA-DR	*IL1R1 IL23R RORC AHR NCR2*	[202]
NK	mNK	Mouse	Blood	CD3^−^CD19^−^CD45.2NK1.1^+^NKp46^+^	*Hbb-bs Smad7 Tgfb1 Qrfp Ifngr1*	[209]
	mNK_Bl1		Blood	CD3^−^CD19^−^CD45.2NK1.1^+^NKp46^+^	*Gzmb Klrg1 Ly6c2 Cma1 Klra9 Ncr1 Emp3 Fgl2*	[209]
	mNK_Bl2		Blood	CD3^−^CD19^−^CD45.2NK1.1^+^NKp46^+^	*Ctla2a Emb Ccr2 Socs3 Xlc1 Cd27 Cd7 Ltb*	[209]
	General		Blood	CD45^+^Lin^−^NK1.1^+^NKp46^+^	*Ly6c2 Klrg1 Klra4*	[210]
	NK (non-tissue- specific)		Blood, spleen, inguinal lymph node, liver, VAT, small intestine IEL/LPL, salivary gland and uterus	CD45^+^Lin^−^NK1.1^+^NKp46^+^	*Itgam S1pr5 Cma1 Zeb2*	[210]
	General		Inguinal lymph node	CD45^+^Lin^−^NK1.1^+^NKp46^+^	*Il7r Ly6c2 Klra8 Klra4 Il18r1*	[210]
	General		Liver	CD45^+^Lin^−^NK1.1^+^NKp46^+^	*Itga1*	[210]
	NK (tissue-specific)		Liver, VAT, salivary gland, uterus, small intestine IEL/LPL and tumor	CD45^+^Lin^−^NK1.1^+^NKp46^+^	*Kit Itga1 Cd160 Asb2 S100a4m Fgl2 Gzmb Cd7m Ctla2 CCr2 Itgb1 Capg Tnfrsf9 Anxa2 Ldha Irf8*	[210]
	NK (early Pec)		Peritoneal exudates	CD3ε^−^NKp46^+^CD49b^+^	*Furin Ctla2a, Ifitm1 Ifng*	[211]
	NK (late Pec)		Peritoneal exudates	CD3ε^−^NKp46^+^CD49b^+^	*Spp1 Gzmb Lag3 Ly6a*	[211]
	General		Salivary gland and uterus	CD45^+^Lin^−^NK1.1^+^NKp46^+^	*Klra4 Klra8 Itga1 Il7r*	[210]
	General		*Salmonella*-infected ceca of WT or Rorasg/sg BM-transplanted (BMT) chimeric mice	CD45.2^+^Lin2CD90^+^	*Ncr1 Sell Eomes Gzma*	[201]
	General		Small-intestine IEL/LPL	CD45^+^Lin^−^NK1.1^+^NKp46^+^	*Kit Itga1 Il7r*	[210]
	mNK		Spleen	CD3^−^CD19^−^CD45.2NK1.1^+^NKp46^+^	*Jun Ccl3 Fos Ccl4 Fosb Klf2*	[209]
	mNK_Sp1		Spleen	CD3^−^CD19^−^CD45.2NK1.1^+^NKp46^+^	*Lgals1 Cma1 Gzmb Fgl2 Klra9 Lys6c2 Irf8 Klrg1*	[209]
	mNK_Sp2		Spleen	CD3^−^CD19^−^CD45.2NK1.1^+^NKp46^+^	*Ctla2a Ltb Emb Cd28 Xcl1 Cd7 Spry2 Fosb Cd27 Cebpb*	[209]
	mNK_Sp3		Spleen	CD3^−^CD19^−^CD45.2NK1.1^+^NKp46^+^	*Nfkbia Nr4a1 Pim1 Prr7 Ccl4*	[209]
	General		Spleen	CD45^+^Lin^−^NK1.1^+^NKp46^+^	*Ly6c2 Klrg1 Klra4 Klra8 Il18r1*	[210]
	General		Spleen	CD3^−^CD19^−^NK1.1^+^NKp46^+^Eomes-GFP^+^CD49a^−^	*Eomes Itgam Zeb2 (no expression of Cd27 Itga1)*	[212]
	NK (early Spl)		Spleen	CD3ε^−^NKp46^+^CD49b^+^	*Xcl1 Ncr1 Jun Fos*	[211]
	NK (resting Spl)		Spleen	CD3ε^−^NKp46^+^CD49b^+^	*Cma1 Klf2 Zeb2 Itga4 Klrc2*	[211]
	NK (proliferating Pec/Spl)		Spleen/peritoneal exudates	CD3ε^−^NKp46^+^CD49b^+^	*Mki67 Top2a Stmn1*	[211]
	Cd11bˡᵒʷCd27ʰᶦᵍʰ		TME	CD3ε^−^NK1.1^+^	*Kit Pdcd1 Tigit Ctla4*	[213]
	Cd11bʰᶦᵍʰCd27ˡᵒʷ		TME	CD3ε^−^NK1.1^+^	*Klrg1 Ncr1 Klrb1c Itga2 Klra3 Klra9 Prf1 Gzma Gzmb*	[213]
	NK(Hif1a−/−)		TME	CD3ε^−^NK1.1^+^	*Ifng Cd69 Ccl4 Ccl3 Nr4a1*	[213]
	General		VAT	CD45^+^Lin^−^NK1.1^+^NKp46^+^	*Itga1 Il7r Ly6c2*	[210]
	hNK	Human	Blood	CD3^−^CD14^−^CD19^−^CD45^+^CD56^low/+^	*PTMA S100A6 TGFB1 GNLY LDHA SCL75A*	[209]
	hNK_Bl1		Blood	anti-CD3, -CD14, -CD19, -CD45, -CD56	*FGFBP2 GZMB GZMA SPON2 CST7 FGCR3A GTF3C1*	[209]
	hNK_Bl2		Blood	anti-CD3, -CD14, -CD19, -CD45, -CD56	*GZMK CD44 CXCR3 SCML1 NCF1 XCL1 SCML1 ZFP36L2*	[209]
	General		Blood	CD45^+^CD56^+^CD3ε^−^CD4^−^CD8a^−^CD14^−^CD15^−^CD163^−^	*GZMK JUNB LTB LNFS*	[214]
	bNK0		Blood	CD45^+^CD56^+^CD3ε^−^CD4^−^CD8a^−^CD14^−^CD15^−^CD163^−^	*FCGR3A FCGR3B PRF1 GZMB*	[214]
	bNK1		Blood	CD45^+^CD56^+^CD3ε^−^CD4^−^CD8a^−^CD14^−^CD15^−^CD163^−^	*GZMK SELL*	[214]
	bNK2		Blood	CD45^+^CD56^+^CD3ε^−^CD4^−^CD8a^−^CD14^–^CD15^−^CD163^−^	*SELL IL7R XCL1/2*	[214]
	bNK3		Blood	CD45^+^CD56^+^CD3ε^−^CD4^−^CD8a^−^CD14^−^CD15^−^CD163^−^	*PCNA MKI67*	[214]
	h_NK_Bm1		BM	CD3^−^CD14^−^CD19^−^CD45^+^CD56^low/+^	*FGFBP2 GZMB GZMH PRF1 FCGR3A*	[215]
	h_NK_Bm2		BM	CD3^−^CD14^−^CD19^−^CD45^+^CD56^low/+^	*CCL3 CCL4 XCL1 XCL2 GZMK CCL3L1 AREG CD160 CD69*	[215]
	h_NK_Bm3		BM	CD3^−^CD14^−^CD19^−^CD45^+^CD56^low/+^	*GZMK XCL1 XCL2 AREG LTB SELL CD44*	[215]
	h_NK_Bm4		BM	CD3^−^CD14^−^CD19^−^CD45^+^CD56^low/+^	*CCL5 GZMM GZMH ZEB2*	[215]
	General		BM	Not provided	*GZMH GZMK GNLY*	[216]
	General		Bronchoalveolar lavage fluid	Not provided	*TYROBP KLRD1 NKG7 FCER1G GNLY PRF1 GZMB KLRF1 XCL1/2 CCL3/4*	[217]
	General		Liver/cirrhotic liver	CD45^+^	*KLRF1 CCL3 XCL1 IL2RB XCL2 CD160 KDLR1 CLEBPD CLIC3 LAT2 CCL3L3*	[218]
	cNK		Liver/cirrhotic liver	CD45^+^	*KLRF1 GNLY PRF1 FGFBP3 SPON2*	[218]
	General		Liver	Viable cells	*NKG7 CCL4 CCL5 KLRD1 KLRK1 GZMA GZMB GZMH*	[219]
	General		Liver	Viable cells	*CD7 KLRB1 NKG7*	[220]
	trNK		Liver	CD45^+^	*KLRB1 TRDC CXCR6 EOMES GZMK KLRF1*	[221]
	cNK		Liver	CD45^+^	*TBX21 CX3CR1 GZMB TRDC FGCR3A KLRF1*	[221]
	NK-GZMH		Intrahepatic cholangiocarcinoma	Viable cells	*KLRF1 KLRB1 IL7R LTB CCR6 NCR3*	[222]
	NK-GZMK		Intrahepatic cholangiocarcinoma	Viable cells	*KLRF1 TYROBP FCER1G GZMK CD160*	[222]
	General		Melanoma metastasis	CD45^+^CD56^+^CD3ε^−^CD4^−^CD8a^−^CD14^−^CD15CD163^−^	*FGFBP2 FCGR3A PRF1 S1PR5 GZMB*	[214]
	tNK0		Melanoma metastasis	CD45^+^CD56^+^CD3ε^−^CD4^−^CD8a^−^CD14^−^CD15^−^CD163^−^	*AREG SELL XCL1/2 FOS*	[214]
	tNK1		Melanoma metastasis	CD45^+^CD56^+^CD3ε^−^CD4^−^CD8a^−^CD14^−^CD15^−^CD163^−^	*CCL3/4L2/5 CD160 TIGIT*	[214]
	tNK2		Melanoma metastasis	CD45^+^CD56^+^CD3ε^−^CD4^−^CD8a^−^CD14^−^CD15^−^CD163^−^	*FGFBP2 FGCR3A PRF1 GZMB*	[214]
	tNK3		Melanoma metastasis	CD45^+^CD56^+^CD3ε^−^CD4^−^CD8a^−^CD14^−^CD15^−^CD163^−^	*SELL SPTSSB KIR2DL4*	[214]
	tNK4		Melanoma metastasis	CD45^+^CD56^+^CD3ε^−^CD4^−^CD8a^−^CD14^−^CD15^−^CD16^−^	*LST1 LTB*	[214]
	tNK5		Melanoma metastasis	CD45^+^CD56^+^CD3ε^−^CD4^−^CD8a^−^CD14^−^CD15^−^CD163^−^	*CCL5 HLA-DRBA GZMH*	[214]
	tNK6		Melanoma metastasis	CD45^+^CD56^+^CD3ε^−^CD4^−^CD8a^−^CD14^−^CD15^−^CD163^−^	*PCNA MKI67*	[214]
	tNK7		Melanoma metastasis	CD45^+^CD56^+^CD3ε^−^CD4^−^CD8a^−^CD14^−^CD15^−^CD163^−^	*ISG15 IFI6*	[214]
	General		Ovarian cancer ascites	CD45^+^	*KLRB1 KLRF1*	[223]
	General		Ovarian cancer ascites	CD45^+^	*FCGR3A FCGR3B NCAM1 KLRB1 KLRB1 KLRC1 KLRD1 KLRF1 KLRK1*	[224]
	General		PBMC	Not provided	*KLRF1 FCGR3A*	[225]
	General		PBMC	Not provided	*FCGR3A NCAM1 GZMB*	[226]
	General		PBMC	Not provided	*KLRF1*	[227]
	hNK		Spleen	CD3^−^CD14^−^CD19^−^CD45^+^CD56^low/+^	*KLF6 NFKBIA TSC22D3 ADGRE5 NCL1 CD69 ANXA1*	[209]
	hNK_Sp1		Spleen	CD3^−^CD14^−^CD19^−^CD45^+^CD56^low/+^	*FGFBP2 GZMB CST7 FGCR3A MYOM2 GNLY S100A4 CYBA PRF1*	[209]
	hNK_Sp2		Spleen	CD3^−^CD14^−^CD19^−^CD45^+^CD56^low/+^	*GZMK XCL1 COTL1 CD160 TOX2 RGS1 LIF KLRB1 KLRB1 BCL2A1 SPRY2*	[209]
	hNK_Sp3		Spleen	CD3^−^CD14^−^CD19^−^CD45^+^CD56^low/+^	*IL7R DUSP4 CD52 GRP183 SELL CD44 CAPG LTB YBX3*	[209]
	hNK_Sp4		Spleen	CD3^−^CD14^−^CD19^−^CD45^+^CD56^low/+^	*TTC30B TXNIP ARRDC3 SDF2 INPP5F*	[209]
	General		Tonsil	CD45^+^Lin^−^CD127^−^CD56^+^NKG2A^+^	*KLRC1 GNLY GZMA EOMES*	[202]
	General		TME/PBMC	CD3CD4CD8CD25	*CD160 XCL1 XCL2 MYADAM CAPG RORA NR4A1/2/3 KLRC1/2/3 IKZF2 EDNTPD1 CD69 ITGAE*	[228]
	NK-1		Tumor/blood	CD3CD19CD20^−^	*IFIT1/3 IFI6 CD8a*	[229]
				CD14^−^CD34^−^CD68^−^CD56^+^/^−^CD127^+^/^−^		
	NK-2		Tumor/blood	CD3^−^CD19^−^CD20^−^CD14^−^CD34^−^CD68^−^CD56^+^/^−^CD127^+^/^−^	*NR4A2 REL CXCR4*	[229]
	NK-ILC1		Tumor/blood	CD3^−^CD19^−^CD20^−^CD14^−^CD34^−^CD68^−^CD56^+^/^−^CD127^+^/^−^	*CD83 IL7R SELL*	[229]

^1^ When provided and different from cell population (general). ^2^ Selected on the basis of the relevance in the study.

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
