# Peer review of "The Dual Role of Innate Lymphoid and Natural Killer Cells in Cancer. from Phenotype to Single-Cell Transcriptomics, Functions and Clinical Uses"

_cancers, 2021, doi:10.3390/cancers13205042_

Round 1

Reviewer 1 Report

The authors of the manuscript titled “Innate Lymphoid and Natural Killer Cells in Cancer. From Phe-notype to Single-cell Transcriptomics, Function and Clinical Use” have summarized the function and impact of the different ILC subsets in cancer prognosis. Furthermore, they have described scRNAseq datasets related to ILCs in tumors and non-tumoral tissues. While interesting, this manuscript lack of overall clarity and coherence. This article must be highly edited and proofread by an native English speaker before acceptance. There are many typos both in the core of the manuscript and in figures (i.e. Figure 1: CYTOLYTIC, EOSINOPHIL) that need to be corrected. The authors have missed many critical studies and some of them are listed below and need to be included in the revised version of the manuscript. In addition, below are some comments and suggestions aiming to improve the content of this article.

Section 2.1.

  • Please, mention that you are describing human circulating NK cells. In lymph nodes and liver, the proportion of CD56 bright and CD56 dim are very different (inversed). Example: DOI: 10.1016/j.cell.2020.01.022
  • The CD56dim CD16- are well known NK cells that have been described a long time ago. These are target/antigen-activated NK cells that have downregulated CD16 upon engagement of this FC receptor. Also, CD16 shedding by the metalloproteases is a well-known mechanism. Please, revise this part by including the relevant information and references. These are few examples: doi: 10.1038/sj.leu.2404499; doi: 10.1182/blood-2012-04-425397; https://doi.org/10.1182/blood-2010-09-303057.
  • Some references are out-dated. Can you please use the most recent reviews where key concepts are introduced. Example – NK cell receptors. Also it would be interesting to quote the seminal papers from Alexandro and Lorenzo Moretta.

Section 2.2

  • Link between lung cancer and NK cells. NK cells may also harbour a regulatory phenotype, influencing disease outcomes (i.e PMID: 30713781).
  • A large body of literature is missing – work from Anne Caignard et al.; work from Laurence Zitvogel et al., in melanoma, lung cancer, GIST, high risk neuroblastoma. In addition, some important references are missing such as https://doi.org/10.3389/fimmu.2020.01242; doi: 10.1158/2326-6066.CIR-18-0500;

Section 2.3

  • Please, quote this article to support the discovery of ILCp in peripheral blood: DOI: 10.1016/j.cell.2017.02.021.

Section 2.4.1

  • Reference 74 may be wrong. Also, this recently published article supports the transdifferentiation of ILC3 into ILC1 in colorectal tumors: DOI: 1016/j.cell.2021.07.029

Section 2.4.2

  • First paragraph, reference 75 may be wrong. You perhaps refer to this study: DOI: 1172/JCI89717 (reference 79)
  • Recent evidence suggest that ILC2 may promote anti-tumor responses: doi: 10.1038/s41590-021-00943-z.; doi: 10.3390/cancers13030559.; doi: 10.1038/s41586-020-2015-4.; DOI: 1016/j.celrep.2020.01.103 . This should be discussed.
  • These references are missing: doi: 10.1038/s41590-020-0745-y and DOI: 1038/s41422-020-0312-y
  • ILC2-derived IL-9 production may also promote anti-tumor responses. Please discuss this point: doi: 10.1016/j.canlet.2021.01.002.

Section 2.4.3

  • Please, discuss these research articles: DOI: 1038/s41422-020-0312-y ; doi: 10.1084/jem.20162031. doi: 10.1158/2326-6066.CIR-19-0775. doi: 10.1016/j.cell.2021.07.029.
  • Please discuss a potential link between ILC3 – GM-CSF - Eosinophils (Fiona Powrie work (ILC3) and Ariel Munitz work (eosinophils) in the intestine/colorectal cancer – Also the work from Mortha A et al. Science 2014 (DOI: 1126/science.1249288 ) should be discussed (GM-CSF expression in ILC3)) and anti-tumor responses in colorectal cancer.

Section 3.1

  • Please check reference 109 as it seems to have an issue with the citation reference manager.
  • Reference 118 seems to be wrong

Section 3.3

  • Please make sure to distinguish human and mouse IgG isotypes. In human, IgG1 is the most potent ADCC inducer while in mouse, it is the IgG2a isotype (please see the work from Pierre Bruhns and colleagues on Fc receptors)

Section 4.1

  • Please don’t hesitate to quote comprehensive reviews in the field to support your claims such as https://doi.org/10.3389/fimmu.2018.02582
  • Please, remove the sentence “ However, some challenges, such as the identification of cell populations, for which a gene expression profile is unknown, remain an issue.” as the primary use of scRNAseq is to identify new populations/cell types whom the transcriptional profile, by definition, is unknown. Few examples include: DOI: 1038/s41586-019-1373-2 ; DOI: 10.1038/s41586-020-2922-4 ; DOI: 10.1038/s41586-020-2157-4 ; doi: 10.1038/s41586-018-0393-7;  DOI: 10.1038/s41586-018-0394-6

Section 4.2

  • Paragraph on ILC3, please add this work: doi: 10.1016/j.cell.2021.07.029
  • When referring to PD-1 expression on ILC progenitors, can you please quote this reference: doi: 10.1016/j.celrep.2016.09.025.

I don’t find the section 4 very informative. The author claim that they provide a comprehensive overview of the scRNAseq studies in the ILC field but many seminal research articles are not described and some organs are completely missed (i.e. liver: https://doi.org/10.3389/fimmu.2021.649311). Please revise accordingly.

Author Response

# Reviewer 1 (see the attachment for color version)

The authors of the manuscript titled “Innate Lymphoid and Natural Killer Cells in Cancer. From Phe-notype to Single-cell Transcriptomics, Function and Clinical Use” have summarized the function and impact of the different ILC subsets in cancer prognosis. Furthermore, they have described scRNAseq datasets related to ILCs in tumors and non-tumoral tissues. While interesting, this manuscript lack of overall clarity and coherence. This article must be highly edited and proofread by an native English speaker before acceptance. There are many typos both in the core of the manuscript and in figures (i.e. Figure 1: CYTOLYTIC, EOSINOPHIL) that need to be corrected. The authors have missed many critical studies and some of them are listed below and need to be included in the revised version of the manuscript. In addition, below are some comments and suggestions aiming to improve the content of this article.

A: We thank the reviewer for her/his detailed comments. We added most of the references suggested by the reviewer. We corrected the typos in the figure and in the main text of the manuscript. A native English speaker checked the final version of the revised manuscript. Below the reviewer could find the detailed answers to all the comments.

Section 2.1.

  • Please, mention that you are describing human circulating NK cells. In lymph nodes and liver, the proportion of CD56 bright and CD56 dim are very different (inversed). Example: DOI: 10.1016/j.cell.2020.01.022

A: We thank the reviewer for the comment and we now specify ‘circulating’ when mentioning described NK cells in the paragraph.

  • The CD56dim CD16- are well known NK cells that have been described a long time ago. These are target/antigen-activated NK cells that have downregulated CD16 upon engagement of this FC receptor. Also, CD16 shedding by the metalloproteases is a well-known mechanism. Please, revise this part by including the relevant information and references. These are few examples: doi: 10.1038/sj.leu.2404499; doi: 10.1182/blood-2012-04-425397; https://doi.org/10.1182/blood-2010-09-303057.

A:   We thank the reviewer for this clarification. We corrected, as suggested, and added citations from 2005, 2007 and 2008 describing CD56dim CD16- NKs. Furthermore, we added information about the regulation of CD16 expression by metalloproteases as indicated by the reviewer. Finally, we extended the explanation about CD56dim CD16- NKs in hematologic malignancies.

Some references are out-dated. Can you please use the most recent reviews where key concepts are introduced. Example – NK cell receptors. Also it would be interesting to quote the seminal papers from Alexandro and Lorenzo Moretta.

A: We updated the references by citing more recent reviews in this section, including the studies of Alessandro and Lorenzo Moretta. 

Section 2.2

  • Link between lung cancer and NK cells. NK cells may also harbour a regulatory phenotype, influencing disease outcomes (i.e PMID: 30713781).

A: We described the role of circulating Nkp46+ NK subset in NSCLC taking into consideration and citing the suggested study.

  • A large body of literature is missing – work from Anne Caignard et al.; work from Laurence Zitvogel et al., in melanoma, lung cancer, GIST, high risk neuroblastoma. In addition, some important references are missing such as https://doi.org/10.3389/fimmu.2020.01242; doi: 10.1158/2326-6066.CIR-18-0500;

A: We thank the reviewer for the comment; we added the required citations and discussed them briefly in the manuscript, thus integrating all the missing parts of literature.

Section 2.3

  • Please, quote this article to support the discovery of ILCp in peripheral blood: DOI: 10.1016/j.cell.2017.02.021.

A: We thank the reviewer for the comment, we added this citation again; unfortunately, it was deleted by a typo in the submission process of the manuscript.

Section 2.4.1

  • Reference 74 may be wrong. Also, this recently published article supports the transdifferentiation of ILC3 into ILC1 in colorectal tumors: DOI: 1016/j.cell.2021.07.029

A: We thank the reviewer for this observation; we provided the required citations and corrected the wrong one.

Section 2.4.2

  • First paragraph, reference 75 may be wrong. You perhaps refer to this study: DOI: 1172/JCI89717 (reference 79)

A: We corrected the refusal by mentioning the correct citation.

  • Recent evidence suggest that ILC2 may promote anti-tumor responses: doi: 10.1038/s41590-021-00943-z.; doi: 10.3390/cancers13030559.; doi: 10.1038/s41586-020-2015-4.; DOI: 1016/j.celrep.2020.01.103 . This should be discussed.

A: We discussed these studies in the indicated section.

  • These references are missing: doi: 10.1038/s41590-020-0745-y and DOI: 1038/s41422-020-0312-y

A: We added the first study in this section. While the second was already present in the section 4.1 and now added also in the section 2.4.3.

  • ILC2-derived IL-9 production may also promote anti-tumor responses. Please discuss this point: doi: 10.1016/j.canlet.2021.01.002.

A: We discussed these studies in the indicated section.

Section 2.4.3

  • Please, discuss these research articles: DOI: 1038/s41422-020-0312-y ; doi: 10.1084/jem.20162031. doi: 10.1158/2326-6066.CIR-19-0775. doi: 10.1016/j.cell.2021.07.029.

A: We discussed and cited the articles suggested and we think that these improved the review since they provide a wider view of the role of ILCs in cancer. The study doi: 10.1084/jem.20162031  was added in the section 4.2, when we describe the relevance of tissue niche in  creating a microenvironment that promotes ILC diversity. Indeed, this research describes the role of the microenvironment in defining ILC fate rather than the ILC3 role in cancer. Finally, we would like to mention that our manuscript was submitted at the end of August 2021 and that one of the papers suggested by the reviewer was published on the 17th of August 2021, when the first draft and the structure of the review was already decided. However, as required, we inserted it in the revised manuscript, aware of its importance.

Please discuss a potential link between ILC3 – GM-CSF - Eosinophils (Fiona Powrie work (ILC3) and Ariel Munitz work (eosinophils) in the intestine/colorectal cancer – Also the work from Mortha A et al. Science 2014 (DOI: 1126/science.1249288 ) should be discussed (GM-CSF expression in ILC3)) and anti-tumor responses in colorectal cancer.

A:  We added a paragraph discussing the research of F. Powrie (DOI: 10.7554/eLife.10066) and Mortha et al. (DOI: 1126/science.1249288 ). However, we did not find any article of A. Munitz that links GM-CSF eosinophils CRC and ILC3s, therefore we were not able to include it here. Nevertheless, we included an explanation of eosinophils recruitment in section 2.4.2 by citing two articles from Munitz (https://doi.org/10.1038/s41568-020-0283-9, DOI: 0.1158/2326-6066.CIR-18-0494): one describing the role of eosinophils in response to IL-5 produced by ILCs and the other the role of eosinophils in cancer.

Section 3.1

  • Please check reference 109 as it seems to have an issue with the citation reference manager.

A:  We provided the correct format for the reference.

  • Reference 118 seems to be wrong

A:  We cited the correct article DOI: 10.1158/2326-6066.CIR-15-0118. The number has changed accordingly to the addition of the new references.

Section 3.3

  • Please make sure to distinguish human and mouse IgG isotypes. In human, IgG1 is the most potent ADCC inducer while in mouse, it is the IgG2a isotype (please see the work from Pierre Bruhns and colleagues on Fc receptors)

 A: We thank the reviewer for this observation. This section deals with clinical applications, so for the sake of brevity, a discussion of the human vs murine isotype seems to us beyond the scope of the manuscript. In any case, we would like to leave to the Editor the final word about this issue.

Section 4.1

  • Please don’t hesitate to quote comprehensive reviews in the field to support your claims such as https://doi.org/10.3389/fimmu.2018.02582

A:  We thank the reviewer for the comment and we added this citation.

  • Please, remove the sentence “ However, some challenges, such as the identification of cell populations, for which a gene expression profile is unknown, remain an issue.” as the primary use of scRNAseq is to identify new populations/cell types whom the transcriptional profile, by definition, is unknown. Few examples include: DOI: 1038/s41586-019-1373-2 ; DOI: 10.1038/s41586-020-2922-4 ; DOI: 10.1038/s41586-020-2157-4 ; doi: 10.1038/s41586-018-0393-7; DOI: 10.1038/s41586-018-0394-6

A:  We agree with the reviewer that one of the main aims of scRNA sequencing is to identify new/unknown cell populations by investigating their transcriptome, as we stated along the manuscript. In this sentence we wanted to highlight that cell populations previously identified/classified with surface markers and for which a transcriptional profile is unknown are not always easy to be identified with this technique. Only recently with multi-omics approaches, such as CITE-seq (https://www.nature.com/articles/nmeth.4380) this issue was partially addressed, as also mentioned in some of the studies suggested by the reviewer. However, we agree that the sentence was not clear, therefore we changed it in the following manner: “However the assignment of cell populations previously characterized with surface protein markers and for which a gene expression profile is unknown, remains challenging.”

Section 4.2

  • Paragraph on ILC3, please add this work: doi: 10.1016/j.cell.2021.07.029

A:  In this section, we discussed studies that employed scRNA-seq for the identification of ILC3s with a focus on pathological conditions. The reference to which the reviewer refers does not present scRNA-seq analyses. Indeed, in the “Limitations of the study” section, it mentions that “Deeper analyses with single-cell RNA-sequencing would comprehensively define the evolution of every T cell compartments during homeostasis, cancer, and treatment with immunotherapy”. However, it includes the discussion of some scRNA-seq data citing Björklund et al., 2016 which was present in our manuscript (ref 196 in the original submission, now ref 247). In any case we added the paper suggested (doi: 10.1016/j.cell.2021.07.029) as ref 106 in section 2.4.1.

  • When referring to PD-1 expression on ILC progenitors, can you please quote this reference: doi: 10.1016/j.celrep.2016.09.025.

A: We cited this study as suggested by the reviewer.

  • I don’t find the section 4 very informative. The author claim that they provide a comprehensive overview of the scRNAseq studies in the ILC field but many seminal research articles are not described and some organs are completely missed (i.e. liver: https://doi.org/10.3389/fimmu.2021.649311). Please revise accordingly.

A: We agree with the reviewer that we may have missed some studies that may include scRNA-seq analyses/data on ILCs (including NKs). However this is a very comprehensive summary of relevant scRNA-seq studies with the aim to characterize and identify ILCs (including NKs) with a focus on cancer/pathological conditions. Often researchers struggle in finding gene markers that identify rare / specific cell populations previously characterized with cell surface proteins, making this as one of the most time-consuming task. In this review, we provided this information in an organized and detailed manner indicating each relevant publication.

Some of the studies that the reviewer suggested are for sure important researches for the identification of intrahepatic NK cells but we thought they were not relevant for the purpose of our since they look at NKs from a general perspective. Out of the five studies reported in the suggested review  (https://doi.org/10.3389/fimmu.2021.649311) three of them (doi: https://doi.org/10.1016/j.jhep.2020.05.039, doi: https://doi.org/10.1038/s41467-018-06318-7 and doi: https://doi.org/10.1038/s41586-019-1373-2) identified only one cluster of NKs and did not add any information to the one already present in the text of our manuscript. This is because i) they did not detect any relevant changes in the tissue analysed, ii) the aim of the study was to provide a baseline of the T and NK transcriptomes in the human liver or iii) they analysed this specific cluster in combination with others that did not include only NKs. Nevertheless, we added the information of the used gene markers of the three publications in our table. On the other hand, one of the study (doi: https://doi.org/10.1038/s41421-020-0157-z) identified differences between liver-resident and circulating NKs and therefore we believe this would complete a section of our manuscript. We added the relevant info of this study in both the manuscript and table. Finally, the last study (https://doi.org/10.1038/s41586-019-1631-3) included in the suggested review, was already present in our manuscript (ref 167 in the original submission, now 217).  We thank the reviewer for this since we think it improved the strength of the manuscript.

Reviewer 2 Report

This review covers the role of ILCs and NK cells in association with cancer. A large number of reviews addressing interesting aspects of ILCs and NK cells in tumor-related immunity have been published and the number is growing. Update of advancements to the exiting literature would improve the review. 

One of the limitations of the review is that the major message is not clear. Authors provided data, suggesting involvement of ILCs and NK cells in pro- and anti-tumorigenesis by regulating expression of various cytokines and effector molecules in TME. These literature studies, instead of listing them simply, need to be better organized and critically discussed to connect each other and to suggest hypotheses for possible mechanistic links. Some suggestions or comments are added below.

1. The title of the review is somewhat ambiguous and not clear to pinpoint the message of the review. Addition of some words for more specific definition of the contents, such as "dual-role of ILCs" and so on would be helpful. 

2. Figure1 is not very explicative. Proper explanation in the legend about the role of ILCs in pro- and anti-tumorigenic processes would help for better understanding.  

3. Definitely, single cell sequencing analysis is nowadays an indispensable technique. However, putting the literature studies and the discussion under the section "4. How single cell studies had re-shaped the field and will contribute further" could mislead the focus of this review. Positioning the summary and the discussion of scRNA seq analysis studies in each section of ILCs and NK cells would be more comprehensive. Integrating these new scRNA seq data to the existing data would strengthen the hypotheses. 

4. Authors described the NK cell-based cancer immunotherapy. In addition to the pros and cons of NK cell sources, providing summary of general NK cell-based therapeutic applications would be helpful to comprehend better. 

Author Response

# Reviewer 2 (see the attachment for color version)

This review covers the role of ILCs and NK cells in association with cancer. A large number of reviews addressing interesting aspects of ILCs and NK cells in tumor-related immunity have been published and the number is growing. Update of advancements to the exiting literature would improve the review.

One of the limitations of the review is that the major message is not clear. Authors provided data, suggesting involvement of ILCs and NK cells in pro- and anti-tumorigenesis by regulating expression of various cytokines and effector molecules in TME. These literature studies, instead of listing them simply, need to be better organized and critically discussed to connect each other and to suggest hypotheses for possible mechanistic links. Some suggestions or comments are added below.

A: We thank the reviewer for the comments we addressed them below.

  1. The title of the review is somewhat ambiguous and not clear to pinpoint the message of the review. Addition of some words for more specific definition of the contents, such as "dual-role of ILCs" and so on would be helpful.

A: We changed the title accordingly, we thank the reviewer for this suggestion.

  1. Figure1 is not very explicative. Proper explanation in the legend about the role of ILCs in pro- and anti-tumorigenic processes would help for better understanding.

A: We extended the caption of the figure by describing all the mechanisms illustrated for the ILC role in cancer. Moreover, as suggested by Reviewer 1, we modified the typos in the figure.

  1. Definitely, single cell sequencing analysis is nowadays an indispensable technique. However, putting the literature studies and the discussion under the section "4. How single cell studies had re-shaped the field and will contribute further" could mislead the focus of this review. Positioning the summary and the discussion of scRNA seq analysis studies in each section of ILCs and NK cells would be more comprehensive. Integrating these new scRNA seq data to the existing data would strengthen the hypotheses.

A: We agree with the reviewer that connecting scRNAseq studies with previous sections could have been of potential interest and may have improved this review; this actually was our first idea. However, with this section our aim was the one of providing a resource for future studies that will include scRNAseq analysis on the frame of ILCs (including NK) with a focus on cancer/pathological conditions. Therefore, integrating this section along the manuscript would scatter this information and it would make it really difficult for the readers to have a complete overview of the scRNAseq studies. Lastly, only a few of these studies can be linked to the previous sections for three main reasons: 1) as we reported in the introduction of this section it is not always easy to associate known cell populations (classified with surface markers) with clusters at scRNAseq level; 2) ILC-related scRNA studies with a focus on cancers are really few; 3) clinical researches employing scRNAseq data are still at their infancy. In a future review, with the increase of  scRNA data on ILCs (including NKs), we could integrate this information. In any case, we would like to leave to the Editor the final word about this issue.

  1. Authors described the NK cell-based cancer immunotherapy. In addition to the pros and cons of NK cell sources, providing summary of general NK cell-based therapeutic applications would be helpful to comprehend better.

A: We thank the reviewer for this suggestion. A summary was added to the first paragraph.

Reviewer 3 Report

The manuscript entitled:" Innate Lymphoid and Natural Killer Cells in Cancer. From Phenotype to Single-cell Transcriptomics, Function and Clinical
Use" focused on a systemic revision of literature data about the role of immune system in cancer is well written but requires the implementation of major points to be accepted for the publication

  • In the text, the authors report the role of NR on NK cell's surface. In my opinion, the authors should implement this section by elucidating the molecular pathway affected by the activation of this receptor.
  • In the text, the authors briefly discuss the role of NK in clinical trial. In my opinion, it should be a great point if the authors should underline the most relevant on going clinical trials where this approach is investigated
  • In the text, the authors show the molecular features of ILC1, ILC2 asd ILC3. According to this point, i would reccomend to emphasize the differences about the activation of the cell population. In particular, i would reccomend to stress the molecular pathway activated by these cell types.
  • In the text, the authors identify the main approaches suitable for the application of this approach. In my opinion, an extensive analysis of literature data should be implemented to improve this section.

Author Response

# Reviewer 3 (see attachment for color version)

The manuscript entitled:" Innate Lymphoid and Natural Killer Cells in Cancer. From Phenotype to Single-cell Transcriptomics, Function and Clinical Use" focused on a systemic revision of literature data about the role of immune system in cancer is well written but requires the implementation of major points to be accepted for the publication

  • In the text, the authors report the role of NR on NK cell's surface. In my opinion, the authors should implement this section by elucidating the molecular pathway affected by the activation of this receptor.

A: We thank the reviewer for the comment; we think that introducing an additional part about the molecular pathway of each receptor of NK cells deflects from the main aim of the review, which is to report the role of NK cells and ILCs in cancer. The main purpose of this paragraph was to provide a general overview of the NRs which will be then treated along the manuscript rather than deeply describing their functional mechanisms. However, we added a few lines on the NKG2D and NKG2A receptor molecular pathways together with their three additional references.

  • In the text, the authors briefly discuss the role of NK in clinical trial. In my opinion, it should be a great point if the authors should underline the most relevant on going clinical trials where this approach is investigated

A: We thank the reviewer for this suggestion. A summary was added to the first paragraph of section 3.

  • In the text, the authors show the molecular features of ILC1, ILC2 and ILC3. According to this point, i would recommend to emphasize the differences about the activation of the cell population. In particular, I would recommend to stress the molecular pathway activated by these cell types.

A: We thank the reviewer for her/his suggestion. We included some references in order to describe some of the molecular mechanisms, which are at the base of ILC regulation in non- pathologic and pathologic conditions. For example, we described the role of TGF-ß and IL-15 onto MAPK in terms of NK to ILC1 conversion. We described the influence of TGF-β and IL-23 onto ILC1 and ILC3 in CRC. Moreover, we described how AREG displays its harmful role in terms of cancer progression acting in ILC2. We also provided information about AHR and the molecular mechanism through which ILC3 in TME are detrimental in BC. Relevant references were added along the manuscript.

  • In the text, the authors identify the main approaches suitable for the application of this approach. In my opinion, an extensive analysis of literature data should be implemented to improve this section.

A: Regarding clinical applications, we thank the reviewer for this suggestion. A summary was added to the first paragraph of section 3. Regarding the scRNAseq section, due to the great amount of data generated from the studies presented in this review and the different platforms employed from each of them, together with the time needed to request, download, analyse and interpret even a few portions it will be beyond the scope of writing this review. However, this might be of potential interest for a new research article with the aim of integrating and analysing ILCs (including NKs) of different dataset and in different conditions.   

Round 2

Reviewer 1 Report

I thank the authors for this remarkable tour de force. The manuscript has been critically improved. Some minor edits are still necessary before publication.

Please correct this sentence (page 7, related to reference 124) : "In lung cancer, ...". This study used tumor cells lines injected iv to induce tumor lesions in the lungs. They haven't properly used lung cancer models. Please, modify this sentence to better reflect the experimental settings used in this article. Page 12, section 4.1, first sentence - please correct "Single cell sequencing" into "Single cell RNA sequencing". In table 2, please indicate for each of the study the authors are referring to the origin of cells (mouse or human). Indeed, as the table currently stands, it is difficult to know whether scRNAseq datasets derived from mouse or human tissues.

Some typos and grammar issues still persist (i.e. page 8 - stromal cells). Please make sure to correct them prior to publish the article. In addition, please make sure to use the English or American spelling only. Currently there is a mix of both (i.e. tumor and tumour).

Author Response

Reviewer 1:

I thank the authors for this remarkable tour de force. The manuscript has been critically improved. Some minor edits are still necessary before publication.

  • Please correct this sentence (page 7, related to reference 124) : "In lung cancer, ...". This study used tumor cells lines injected iv to induce tumor lesions in the lungs. They haven't properly used lung cancer models. Please, modify this sentence to better reflect the experimental settings used in this article.

A: We thank the Reviewer for the suggestion. We provided the required correction.

  • Page 12, section 4.1, first sentence - please correct "Single cell sequencing" into "Single cell RNA sequencing".

A: We thank the Reviewer; we provided the required correction.

  • In table 2, please indicate for each of the study the authors are referring to the origin of cells (mouse or human). Indeed, as the table currently stands, it is difficult to know whether scRNAseq datasets derived from mouse or human tissues.

A: We thank the Reviewer for this suggestion. We modified the table inserting dashed lines to better distinguish human/mouse sources. We provided also a better description of the table in the legend.

  • Some typos and grammar issues still persist (i.e. page 8 - stromal cells). Please make sure to correct them prior to publish the article. In addition, please make sure to use the English or American spelling only. Currently there is a mix of both (i.e. tumor and tumour).

A: We thank the Reviewer for highlighting the typos. We corrected them along the manuscript.

Reviewer 2 Report

# Reviewer 2

This review covers the role of ILCs and NK cells in association with cancer. A large number of reviews addressing interesting aspects of ILCs and NK cells in tumor-related immunity have been published and the number is growing. Update of advancements to the exiting literature would improve the review.

One of the limitations of the review is that the major message is not clear. Authors provided data, suggesting involvement of ILCs and NK cells in pro- and anti-tumorigenesis by regulating expression of various cytokines and effector molecules in TME. These literature studies, instead of listing them simply, need to be better organized and critically discussed to connect each other and to suggest hypotheses for possible mechanistic links. Some suggestions or comments are added below.

A: We thank the reviewer for the comments we addressed them below.

  1. The title of the review is somewhat ambiguous and not clear to pinpoint the message of the review. Addition of some words for more specific definition of the contents, such as "dual-role of ILCs" and so on would be helpful.

A: We changed the title accordingly, we thank the reviewer for this suggestion.

-This comment has been addressed fully.

  1. Figure1 is not very explicative. Proper explanation in the legend about the role of ILCs in pro- and anti-tumorigenic processes would help for better understanding.

A: We extended the caption of the figure by describing all the mechanisms illustrated for the ILC role in cancer. Moreover, as suggested by Reviewer 1, we modified the typos in the figure.

-This comment has been addressed sufficiently.

  1. Definitely, single cell sequencing analysis is nowadays an indispensable technique. However, putting the literature studies and the discussion under the section "4. How single cell studies had re-shaped the field and will contribute further" could mislead the focus of this review. Positioning the summary and the discussion of scRNA seq analysis studies in each section of ILCs and NK cells would be more comprehensive. Integrating these new scRNA seq data to the existing data would strengthen the hypotheses.

A: We agree with the reviewer that connecting scRNAseq studies with previous sections could have been of potential interest and may have improved this review; this actually was our first idea. However, with this section our aim was the one of providing a resource for future studies that will include scRNAseq analysis on the frame of ILCs (including NK) with a focus on cancer/pathological conditions. Therefore, integrating this section along the manuscript would scatter this information and it would make it really difficult for the readers to have a complete overview of the scRNAseq studies. Lastly, only a few of these studies can be linked to the previous sections for three main reasons: 1) as we reported in the introduction of this section it is not always easy to associate known cell populations (classified with surface markers) with clusters at scRNAseq level; 2) ILC-related scRNA studies with a focus on cancers are really few; 3) clinical researches employing scRNAseq data are still at their infancy. In a future review, with the increase of  scRNA data on ILCs (including NKs), we could integrate this information. In any case, we would like to leave to the Editor the final word about this issue.

-Explanation of the authors about the aim of the section has addressed the comment sufficiently.  

  1. Authors described the NK cell-based cancer immunotherapy. In addition to the pros and cons of NK cell sources, providing summary of general NK cell-based therapeutic applications would be helpful to comprehend better.

A: We thank the reviewer for this suggestion. A summary was added to the first paragraph.

-This comment has been addressed sufficiently.

Author Response

We thank the Reviewer for useful comments and constructive criticism.

Reviewer 3 Report

The manuscript may be accepted in the present form

Author Response

(The authors gave the same response as above.)
